# In vivo measurement of an Apelin gradient with a genetically encoded APLNR conformation biosensor

Lukas Herdt [1,4], Hannes Schihada[2,4], Michael Kurz [3], Sebastian Ernst[3], Jean Eberlein [1], Peter Kolb [2], Cornelius Krasel [3], Moritz Bünemann [3] & Christian S. M. Helker [1] ✉

The Apelin receptor (APLNR), a class A G-protein coupled receptor, plays a crucial role during cardiovascular development and tumor angiogenesis. To understand its spatiotemporal activity in health and disease is fundamental for the development of drugs to manipulate its activation state. To obtain this understanding, here we develop a tool box of various APLNR conformation biosensors, based on FRET, BRET and the conformation-sensitive fluorophore circularly permuted GFP (cpGFP), with further focus on its in vivo application. We demonstrate the functionality of our biosensors by pharmacological characterization and signal transduction analysis in vitro. Two APLNR-cpGFP biosensors show superior signal-to-noise ratio and are further analyzed for their in vivo applicability. In zebrafish embryos, APLNR-cpGFP biosensors are able to bind both endogenous ligands, Apelin and Apela, and visualize endogenous Aplnr activity in growing blood vessels. Moreover, we are able to measure an Apelin ligand gradient across cellular distances in vivo. Hence, these APLNR conformation biosensors are powerful tools to resolve the spatiotemporal Apelin signaling activity in health and disease.

G-protein-coupled receptors (GPCRs) comprise one of the largest membrane protein receptor families. They transduce extracellular stimuli into the cell interior and feature a characteristic architecture including seven transmembrane-spanning helices and three intra- and extracellular loop regions. GPCRs are involved in a broad variety of physiological and pathophysiological processes. Hence, their pathological implications combined with the accessibility for extracellular molecules render GPCRs a frequently targeted class of receptors with a huge therapeutic potential.

The Apelin receptor (APLNR) is a class A (rhodopsin-like), peptide-binding GPCR identified in 1993[1]. Apelin signaling is involved in several physiological (e.g., mesodermal migration[2,3] and cardiovascular development[4–11]) and pathophysiological (e.g., cancer[9,12–14], diabetes[15–18] and obesity[19–21]) processes. To date, two

peptide ligands of the APLNR are known, Apelin and Apela (also known as Toddler or Elabela). We and others have recently shown that neural-derived Apelin regulates angiogenesis by modulating endothelial cell behavior and metabolism, thus promoting cell migration and blood vessel development[5,7]. In contrast, Apela is essential for heart development[2] as well as angioblast migration toward the embryonic midline during vasculogenesis in the early embryo[4]. Despite the significance of APLNR and its ligands in these and other physiological processes, we still lack knowledge about the spatiotemporal activity of APLNR and how different ligands and their isoforms can activate and modulate APLNR function differently in vivo. For this, we would need advanced pharmacological, in vivo-applicable tools that allow for the assessment of APLNR activity in space and time.

[1]Department of Biology, Animal Cell Biology, Marburg University, Marburg, Germany. [2]Institute of Pharmaceutical Chemistry, Faculty of Pharmacy, Marburg University, Marburg, Germany. [3]Institute of Pharmacology and Clinical Pharmacy, Faculty of Pharmacy, Marburg University, Marburg, Germany. [4]These authors contributed equally: Lukas Herdt, Hannes Schihada. ✉e-mail: christian.helker@biologie.uni-marburg.de

Engineering of optical protein biosensors is a common approach to investigate GPCR function and activity in living cells, and real time. Two major biophysical principles used are fluorescence (FRET[22-24]) and bioluminescence energy transfer (BRET[25-28]). In addition to these energy transfer-based tools, biosensors based on the integration of a single circularly permuted green fluorescent protein (cpGFP) have also been used successfully to monitor protein conformational rearrangements[29-33]. The fluorescence intensity of cpGFP is dependent on its folding state, which – in turn – is controlled by the overall conformation of the protein of interest, e.g., a cpGFP-tagged GPCR[30,31,33]. All three biosensor principles have been extensively used for the development of conformational GPCR biosensors[22,23,26-28,30,31,33-37].

In the present study, we set out to engineer in vivo-applicable APLNR conformational biosensors and therefore employed all three biosensor designs. Initially, we compare the performance of FRET-, BRET-, and cpGFP-based fusion constructs for the generation of APLNR biosensors. Two cpGFP-based biosensors show superior sensitivity and allow us to monitor APLNR activity and to pharmacologically characterize a set of APLNR ligands in model cell lines. Moreover, we demonstrate that our biosensors function in vivo and detect Apelin ligand concentration gradients in mRNA microinjected zebrafish embryos.

## Results

### Comparison of different APLNR conformational biosensor designs

We used a FRET-, BRET-, and cpGFP-based design to develop conformational biosensors for the APLNR. For FRET and BRET biosensors, the blue fluorescent protein mTurquoise2 (mTq2) and small luciferase NanoLuc, respectively, were fused to the C-terminus, which was truncated after amino acid S343. These energy donors were combined with the yellow fluorescent protein mCitrine, which was inserted into one of four different positions in the short intracellular loop 3 (ICL3) of APLNR. The same four positions in ICL3 were targeted to incorporate the cpGFP module flanked by an N-terminal LSSLI and C-terminal NHDQL linker, as previously described[30]. For the cpGFP-based biosensors, the APLNR C-terminus was not truncated. Our design strategy thus resulted in four FRET, four BRET, and four cpGFP biosensors (Fig. 1a–c).

Each of these fusion constructs was then individually transiently expressed in HEK293 cells and treated with 10 μM of the agonist [Pyr[1]]-Apelin-13 (henceforth referred to as "Apelin") in single-cell microscopy (FRET; Fig. 1d) or 96-well microtiter plate experiments (BRET and cpGFP; Fig. 1e, f). All four FRET constructs showed a decrease of about 10% in FRET in response to agonist treatment (Fig. 1d; Supplementary Fig 1a–i) while none of the BRET variants displayed a significant change in emission ratios (Fig. 1e). Interestingly, Apelin binding to the APLNR-FRET biosensors is reversible, since superfusion of buffer without agonist after stimulation increased the FRET ratio toward the basal FRET level again (Supplementary Fig. 1a–d). Similar to the FRET constructs, all four APLNR-cpGFP variants showed a significant change in fluorescence intensity upon Apelin treatment, with the biosensors labeled after amino acids F233 and K235 displaying the highest signal-to-noise ratio (Fig. 1f).

### Pharmacological validation of APLNR-cpGFP biosensors and characterization of APLNR ligands

Because the cpGFP probes are more easily applicable for live-cell imaging studies and have a better signal-to-noise ratio than the FRET biosensors, we focused on the APLNR(F233)-cpGFP and APLNR(K235)-cpGFP biosensors and sought to pharmacologically validate these engineered APLNR variants. We generated HEK293 cells stably expressing one of the two biosensors and recorded the time course of

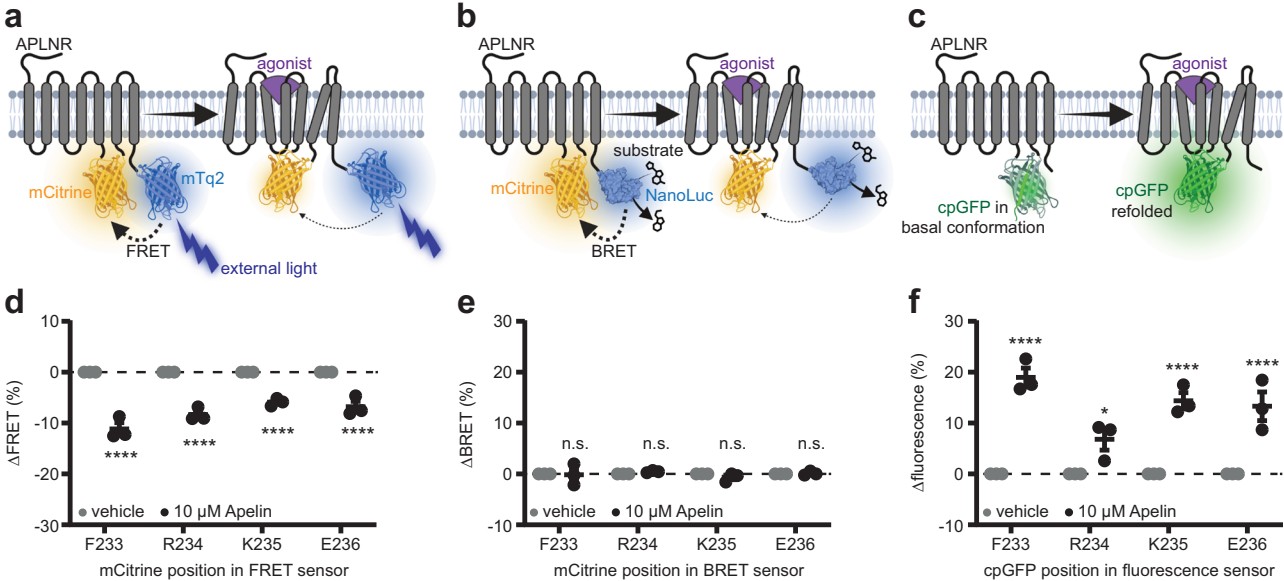

**Fig. 1 | APLNR conformational biosensor design and validation.** Schematic illustration of the APLNR conformational biosensors design based on FRET (**a**), BRET (**b**) and cpGFP (**c**). mCitrine (FRET and BRET) and cpGFP were integrated at four different positions in the intracellular loop domain 3 of the APLNR (F233, R234, K235 and E236). **d**–**f** Functionality validation of the different biosensor variants stimulated with 10 μM Apelin in HEK293 cells. All four FRET biosensor variants show a significant FRET ratio decrease of around 10% after Apelin stimulation (**d**). In contrast, none of the BRET biosensors show a change in the BRET ratio after Apelin stimulation (**e**). All four cpGFP biosensor variants show a significant increase in the biosensor fluorescence intensity after Apelin stimulation (**f**). Data in (**d**) presented as mean ± StD from three transiently transfected single HEK293T cells. Data in (**e**, **f**) are presented as mean values ± SEM from three independent experiments conducted in transiently transfected HEK293A cells. Statistical analysis was performed by using a 2-way ANOVA followed by Šidák's multiple comparison correction (*$p < 0.0332$; ****$p < 0.0001$). mTq2 mTurquoise2, FRET Förster resonance energy transfer, BRET bioluminescence resonance energy transfer, cpGFP circularly permuted GFP, HEK293 human embryonic kidney 293, n.s. not significant. Source data are provided as a Source Data file. **a**–**c** Created in BioRender. Schihada (2025) https://BioRender.com/e05s6eo.

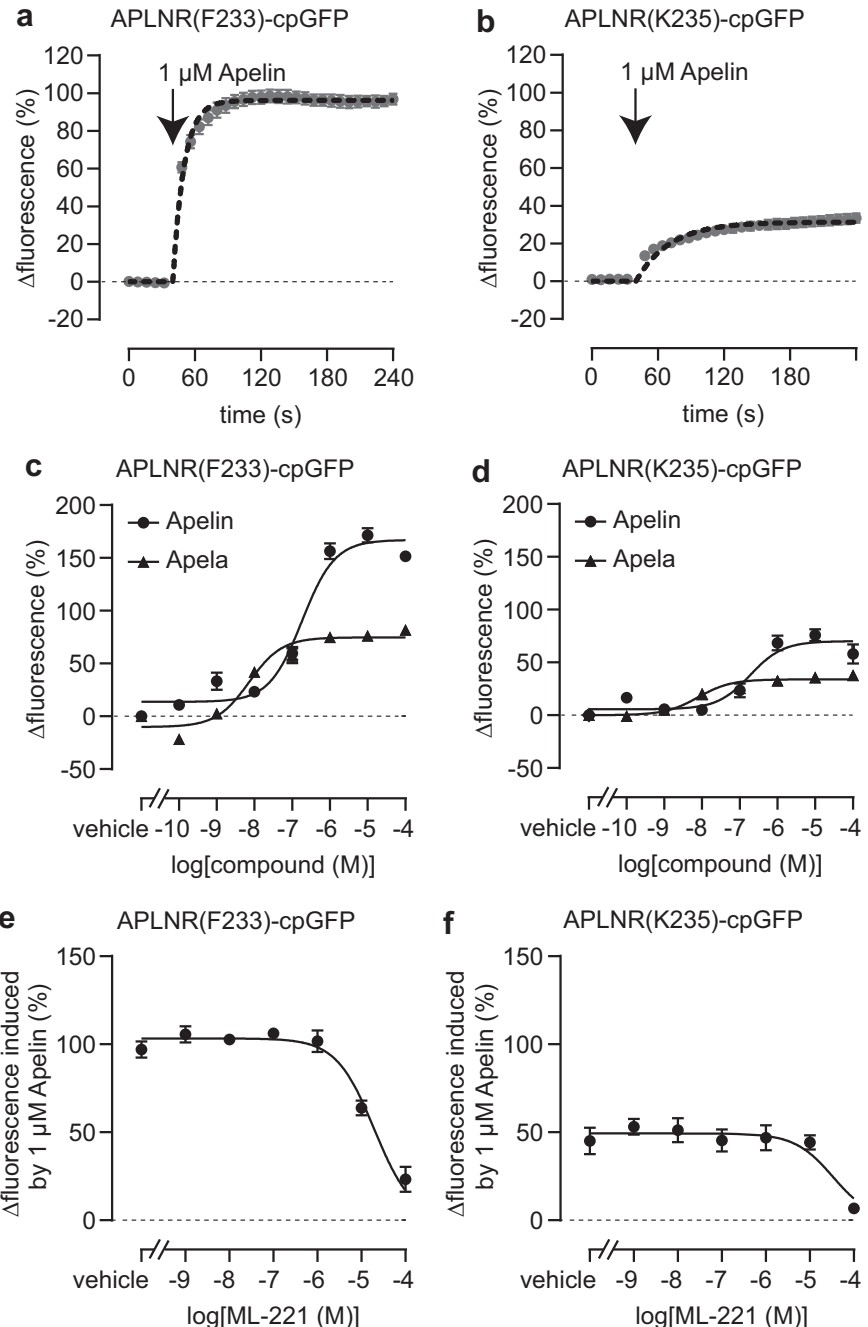

**Fig. 2 | Pharmacological characterization of the APLNR-cpGFP biosensors.** Stimulation with Apelin and Apela (arrow indicates timepoint of ligand addition) led to a significant increase of APLNR(F233)-cpGFP (**a**, **c**) and APLNR(K235)-cpGFP (**b**, **d**) biosensor fluorescent intensity. The APLNR antagonist ML-221 significantly diminished the Apelin-induced fluorescent response of APLNR(F233)-cpGFP (**e**)

and APLNR(K235)-cpGFP (**f**). Data are presented as mean values ± SEM from three independent experiments conducted in stable APLNR-cpGFP expressing HEK293T cells. cpGFP circularly permuted GFP, HEK293T human embryonic kidney 293T. Source data are provided as a Source Data file.

cpGFP emission upon Apelin treatment. Both APLNR-cpGFP biosensors showed an immediate increase in fluorescence upon automated ligand injection into wells cultured with stably expressing HEK293 cells (Fig. 2a, b). In addition, we determined the potencies of the two endogenous ligands, Apelin and Apela, in inducing conformational changes in both variants (Fig. 2c, d). These values (mean $EC_{50}$ [95% confidence interval] of Apelin: 182 nM [112–297 nM] at APLNR(F233)-cpGFP and 186 nM [81–431 nM] at APLNR(K235)-cpGFP; mean $EC_{50}$ [95% confidence interval] of Apela: 7 nM [4–14 nM] at APLNR(F233)-cpGFP and 8 nM [4–15 nM] at APLNR(K235)-cpGFP) were in agreement with previous studies and suggest that Apela is a more

potent but partial agonist of APLNR compared to the full agonist Apelin.

In a competition experiment, we further assessed the activity of ML-221 – a proposed small molecule antagonist of the APLNR[38]. ML-221 blocked the 1 μM Apelin-induced fluorescence response with $IC_{50}$ values around 10–30 μM, yielding $K_i$ values of 3 and 5 μM at APLNR(F233)-cpGFP and APLNR(K235)-cpGFP, respectively (Fig. 2e, f).

**Assessing the signaling capacity of APLNR-cpGFP biosensors**
Conformational GPCR biosensors are powerful tools for studying signaling pathways in living systems. However, it is essential to

understand how effectively these biosensors function when integrated into living organisms. Therefore, we assessed the signaling abilities of APLNR(F233)-cpGFP and APLNR(K235)-cpGFP biosensors for $G_{i1}$ protein activation and Arrestin3 recruitment in HEK293 cells.

Using the heterotrimeric G-protein BRET sensor $G_{i1}$-CASE[39], we confirmed that both APLNR(F233)-cpGFP and APLNR(K235)-cpGFP biosensors, which were expressed at the plasma membrane at levels comparable to two different wildtype APLNR control conditions (Supplementary Fig. 2a), were capable of inducing $G_{i1}$ heterotrimer dissociation upon Apelin stimulation. However, to a lesser extent compared to wildtype APLNR (Fig. 3a; Supplementary Fig. 2b–f). Similarly, a significant but reduced recruitment of Arrestin3 to the plasma membrane was observed when APLNR(F233)-cpGFP and APLNR(K235)-cpGFP were activated by Apelin (Fig. 3b; Supplementary Fig. 3). The reduced ability of both APLNR-cpGFP biosensors to recruit Arrestin3 compared to wildtype APLNR was also observed in live-cell fluorescence imaging experiments (Fig. 3c). While wildtype APLNR-transfected HEK293 cells showed a substantial translocation of mCherry-labeled Arrestin3 to the plasma membrane upon Apelin addition, only a minor membrane localization of fluorescent Arrestin3 was observed in APLNR(F233)-cpGFP or APLNR(K235)-cpGFP transfected cells (Fig. 3c). In conclusion, our results show that both APLNR-cpGFP biosensors are able to promote $G_{i1}$ activation and Arrestin3 recruitment.

### APLNR-cpGFP biosensors are functional in vivo

To determine if our engineered APLNR-cpGFP biosensors exhibit APLNR activity in vivo, we tested the APLNR-cpGFP biosensors in zebrafish embryos. We ubiquitously expressed the *APLNR(K235)-cpGFP* biosensor along with a control *membrane-tomato* marker by injecting their mRNA into one-cell stage zebrafish embryos. Clear membrane localization of the APLNR(K235)-cpGFP biosensor was observed by confocal imaging, confirming correct placement in the cell membrane of our biosensor in vivo. To stimulate the APLNR(K235)-cpGFP in vivo, we used the heat shock-inducible transgenic lines (*Tg(hsp70l:apln)* and *Tg(hsp70l:apela)*) to ubiquitously overexpress *apln* or *apela* (Fig. 4a, b) by a single heat shock at 5.5 h post-fertilization (hpf) and subsequently imaged the embryos 30 min later at 6 hpf by confocal microscopy (Fig. 4b). Stimulation of the APLNR(K235)-cpGFP biosensor by Apelin or Apela in vivo resulted in an increase of APLNR(K235)-cpGFP fluorescence intensities by 27% respectively, indicating that our biosensor responds to ligand activation in vivo (Fig. 4c, d). To exclude the possibility that fluorescence changes are due to positional effects within the embryo, we generated an APLNR-EGFP construct that does not alter GFP fluorescence upon ligand binding. We injected the APLNR-EGFP mRNA into *Tg(hsp70l:apln)* embryos (Supplementary Fig. 4a), induced Apelin expression via heat shock at 5.5 hpf, and imaged the embryos at 6 hpf (Supplementary Fig. 4b). We observed no change in APLNR-EGFP fluorescence intensity (Supplementary Fig. 4c), validating the functionality of the APLNR-cpGFP biosensors and their responsiveness to endogenous Aplnr zebrafish ligands.

To determine if APLNR-cpGFP biosensors can monitor endogenous Aplnr activity in vivo, we created stable zebrafish lines that express the *APLNR(F233)-cpGFP* and *APLNR(K235)-cpGFP* biosensors under the control of the UAS promoter. Since we and others have previously shown that the *aplnr* is predominantly expressed in endothelial cells of the vasculature[4,5,7,9–11], we used a vascular-specific GAL4 driver (*Tg(fli1a:GAL4FF)*; Supplementary Fig. 5) to express our biosensors within endothelial cells (ECs). Based on previous findings that Apelin signaling is required for the formation of intersegmental vessels (ISVs) but not the dorsal aorta (DA) and posterior cardinal vein (PCV)[5,7], we hypothesized that Apelin signaling would be more active in the ISVs than in other vascular beds. To test our hypothesis, we analyzed our transgenic APLNR-cpGFP

biosensors lines at 28 hpf by confocal microscopy and quantified APLNR-cpGFP fluorescence intensities in ECs of the ISVs, DA, and PCV to highlight the relative increase in Aplnr signaling activity in the ISVs compared to the DA/PCV (Fig. 5a, c). Of note, APLNR-cpGFP expression was localized at the cell membrane of ECs, indicating also the correct localization of the transgenic biosensors in vivo. Both APLNR-cpGFP biosensors displayed approximately 50% higher cpGFP fluorescence intensity in ECs of the ISVs compared to those of the DA and PCV (Fig. 5b, d; APLNR(F233)-cpGFP: +47 % and APLNR(K235)-cpGFP: +50%). This increase confirms our hypothesis that Apelin signaling activity is indeed higher in ISV ECs.

To prove that our APLNR-cpGFP biosensors reflect endogenous Aplnr activity in vivo, we validated our APLNR(K235)-cpGFP biosensor in ISVs in *apln*, *apela* double knockout embryos and in Apelin overexpression embryos. The CRISPANT-injected embryos phenocopied the *apln*, *apela* double knockout embryos[5] (Fig. 5e). Knockout of both Aplnr endogenous ligands led to a 35% decrease in APLNR(K235)-cpGFP biosensor intensity in ISVs compared to the control siblings (Fig. 5f). Furthermore, we quantified APLNR(K235)-cpGFP biosensor activity in ISVs in Apelin overexpressing *Tg(hsp70l:apln)* embryos. We induced Apelin overexpression by a single heat shock at 27 hpf and subsequently imaged the embryos at 28 hpf (Fig. 5g). Stimulation by Apelin overexpression resulted in an increase of APLNR(K235)-cpGFP fluorescence intensities by 250% compared to control siblings (Fig. 5h).

In summary, these findings show that our APLNR-cpGFP biosensors can be used to measure Aplnr signaling in vivo and demonstrate their potential as an effective tool for real-time monitoring of APLNR activity in living organisms.

### Measuring an Apelin ligand gradient in vivo

After confirming that our APLNR-cpGFP biosensors function reliably in vivo, we next aimed to determine whether they could measure Apelin ligand gradients within living organisms. To test this, we expressed the *APLNR(K235)-cpGFP* biosensor and a *membrane-tomato* marker by injecting mRNA into one-cell stage zebrafish embryos. We then performed a second injection at the 128-cell stage. We co-injected *apln* mRNA with a Dextran-AlexaFluor647 (Dextran-AF647) tracer to label *apln*-secreting cells (Fig. 6a). Since the *apln* mRNA contains the whole coding sequence, including the signal peptide sequence, we assume the Apelin protein is secreted by these cells. As a control, we injected only the fluorescently labeled Dextran-AF647 without *apln* mRNA (Fig. 6a). Successfully double-injected embryos were sorted on a fluorescent microscope based on cpGFP, membrane-tomato, and Dextran fluorescence, and confocal imaging was performed at 6 hpf (Fig. 6b, d). Our approach assumes that cells closest to *apln*-expressing cells will be exposed to the highest Apelin concentration (cell distance = 1), with this gradient diminishing with increasing cell distance.

We measured the APLNR(K235)-cpGFP biosensor fluorescence of individual cells up to five cells away from Dextran-AF647 positive cells (Fig. 6c, e). In control-injected embryos, the APLNR(K235)-cpGFP biosensor intensity remained constant regardless of a cell's distance from a Dextran-AF647-positive cell (Fig. 6c). In contrast, we observed that cells located closest to an *apln*-expressing Dextran-AF647-positive cell exhibit higher APLNR(K235)-cpGFP biosensor intensities (Fig. 6e). Fluorescence intensity stayed relatively constant within three cell distances but decreased substantially at four and five cell distances, with fluorescence intensity reductions of about 16% and 23%, respectively (Fig. 6e).

These results highlight the power of our generated APLNR(K235)-cpGFP biosensor, demonstrating its sensitivity and applicability for measuring spatially resolved signaling in vivo. By using this cpGFP-based biosensor, we can successfully track Apelin ligand distributions across cellular distances in a living organism.

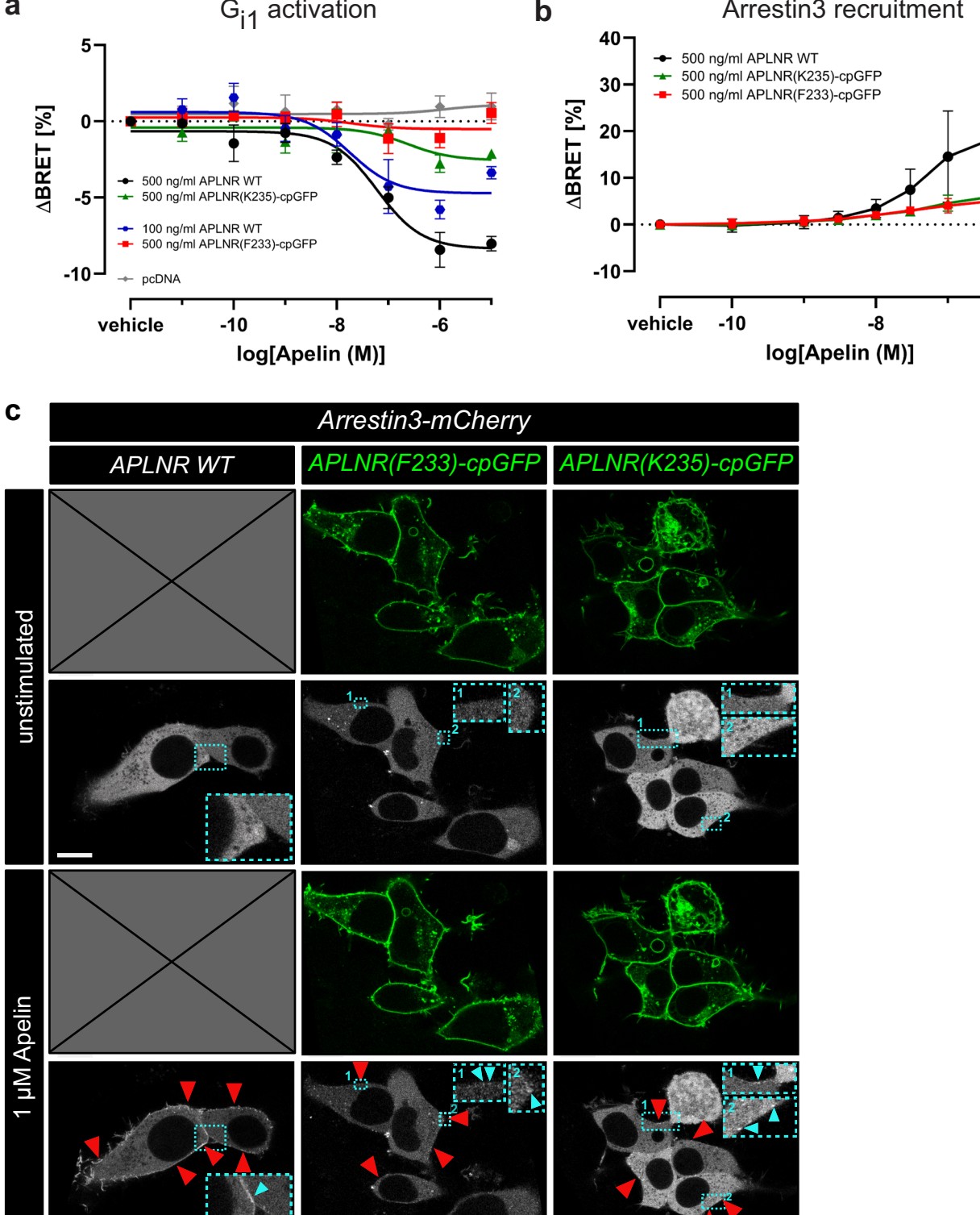

**Fig. 3 | APLNR-cpGFP biosensors possess signaling ability.** $G_{i1}$ protein dissociation (**a**) and Arrestin3 recruitment (**b**) BRET assay of APLNR wildtype (WT), APLNR(F233)-cpGFP, and APLNR(K235)-cpGFP after Apelin stimulation. ng/ml represents the amount of receptor plasmid DNA (ng) used to transfect one ml of cells. **c** HEK293 cells were transiently transfected with Arrestin3-mCherry together with either APLNR WT, APLNR(F233)-cpGFP or APLNR(K235)-cpGFP. Cytoplasmic Arrestin3-mCherry localization is observed in unstimulated conditions. Apelin stimulation leads to a substantial translocation of Arrestin3-mCherry in APLNR WT co-transfected HEK293 cells. In contrast, only a minor fraction of Arrestin3-mCherry is membrane-localized when co-transfected with either APLNR(F233)-cpGFP or APLNR(K235)-cpGFP after Apelin stimulation. Arrowheads indicate membrane-localized Arrestin3-mCherry. Data are presented as mean values ± SEM from three independent experiments (**a**) and ±StD from four independent experiments (**b**) conducted in transiently transfected HEK293A cells (**a**) or transiently transfected HEK293T cells (**b**, **c**). cpGFP circularly permuted GFP, HEK293 human embryonic kidney 293. Source data are provided as a Source Data file.

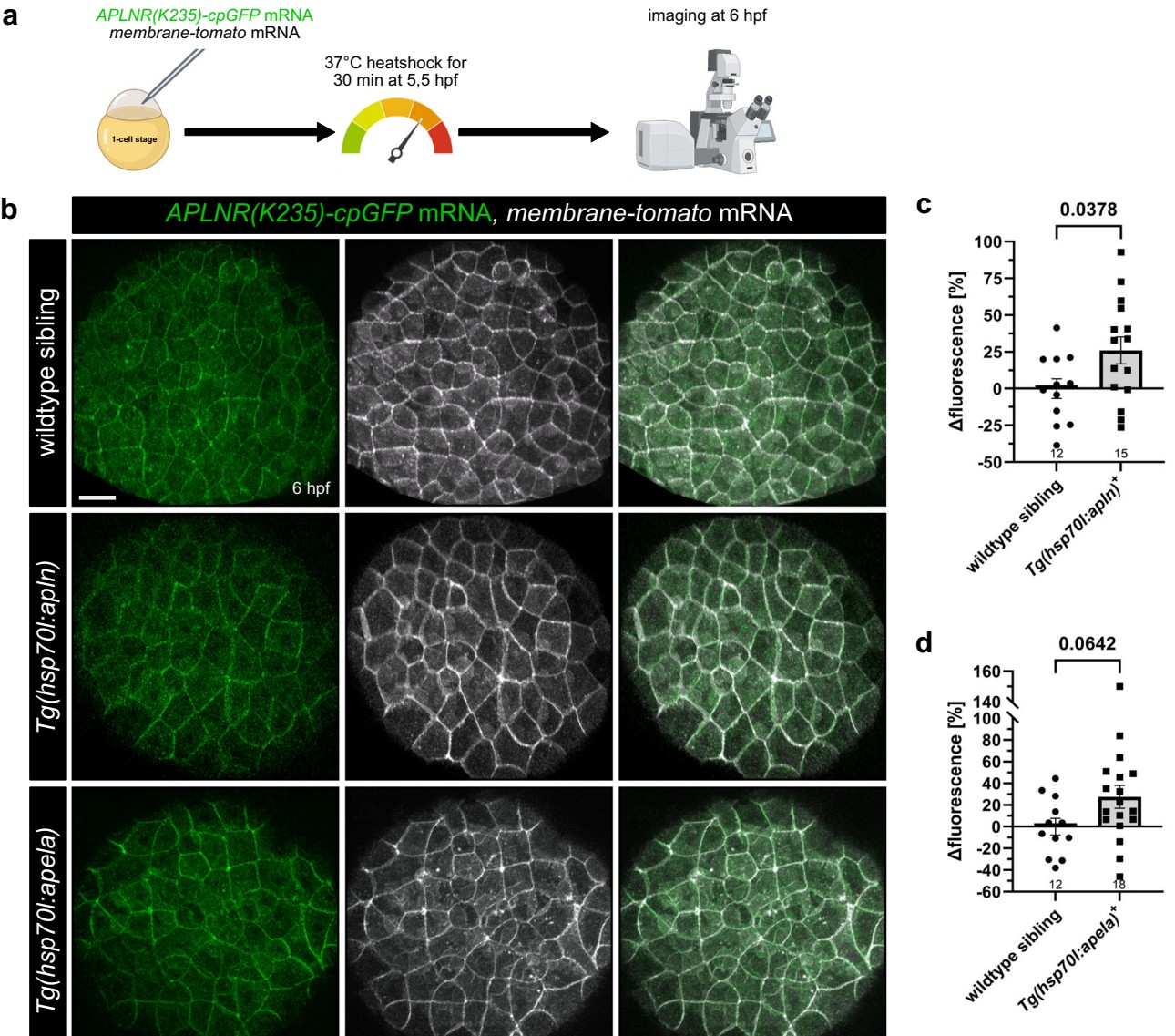

**Fig. 4 | Ubiquitous Apelin and Apela overexpression activate APLNR(K235)-cpGFP biosensor in vivo. a** Schematic illustration of the procedure. *APLNR(K235)-cpGFP* and *membrane-tomato* mRNA were injected into 1-cell stage zebrafish embryos. At 5.5 hours post-fertilization (hpf), injected embryos were heat shocked for 30 min at 37 °C to induce ubiquitous ligand overexpression and subsequently imaged at 6 hpf. **b** Representative confocal projection images of an injected wildtype sibling and transgenic *Tg(hsp70l:apln)* and *Tg(hsp70l:apela)* embryos at 6 hpf, respectively. Quantification of APLNR(K235)-cpGFP delta fluorescence intensity of wildtype siblings compared to embryos ubiquitously overexpressing the *apln* (**c**)

and the *apela* (**d**) ligand. Each dot represents the mean of 10 measured cells per embryo. Data are presented as mean values ± SEM. (*N* number of embryos, *n* number of cells: **c** wildtype siblings *N/n*: 12/120, *Tg(hsp70l:apln)*+ *N/n*: 15/150; **d** wildtype siblings *N/n*: 12/120, *Tg(hsp70l:apela)*+ *N/n*: 18/180). Statistical analysis was performed by using a two-tailed unpaired Student's *t*-test with Welch's correction. Scale bars 30 μm. cpGFP circularly permuted GFP. Source data are provided as a Source Data file. **a** Created in BioRender. Schihada (2025) https://BioRender.com/vqh3y8e.

## Development of ratiometric APLNR-cpGFP-mScarlet-I3 biosensor as an advancement for real-time measurement of the Aplnr activity

To improve the suitability of our cpGFP-based APLNR conformational biosensors for in vivo studies, we developed ratiometric biosensor versions that allow for normalization to distinct biosensor expression levels across tissues and cell types. We fused mScarlet-I3 either directly or via a p2A self-cleaving peptide to the C-terminus of our two best-performing biosensors: APLNR(F233)-cpGFP and APLNR(K235)-cpGFP. This approach yielded four distinct biosensor constructs. The p2A-containing biosensor versions were designed to exclude potential cpGFP quenching due to FRET with mScarlet-I3, which, in the absence of a p2A, remains tethered to APLNR-cpGFP (Fig. 7a). In contrast, the p2A cleavage releases mScarlet-I3 into the cytosol, minimizing the risk

of FRET interference (Fig. 7b). Thus, receptor activation should induce an increase in cpGFP fluorescence without affecting mScarlet-I3, allowing mScarlet-I3 to serve as an internal control for expression levels.

We first validated the four ratiometric biosensors in transiently transfected HEK293A cells (Fig. 7c). Upon Apelin stimulation, all biosensors showed a comparable increase of about 100% in cpGFP over mScarlet-I3 fluorescence intensity (Fig. 7c). We then assessed whether cpGFP and mScarlet-I3 engage in FRET, which could impact cpGFP fluorescence intensities, especially in the biosensor variants without a p2A. No FRET effects were observed, and no differences were detected between the p2A and non-p2A versions (Fig. 7d), confirming that FRET does not impact the ratiometric biosensor signals (Fig. 7d). Given their robust performance in vitro, we next

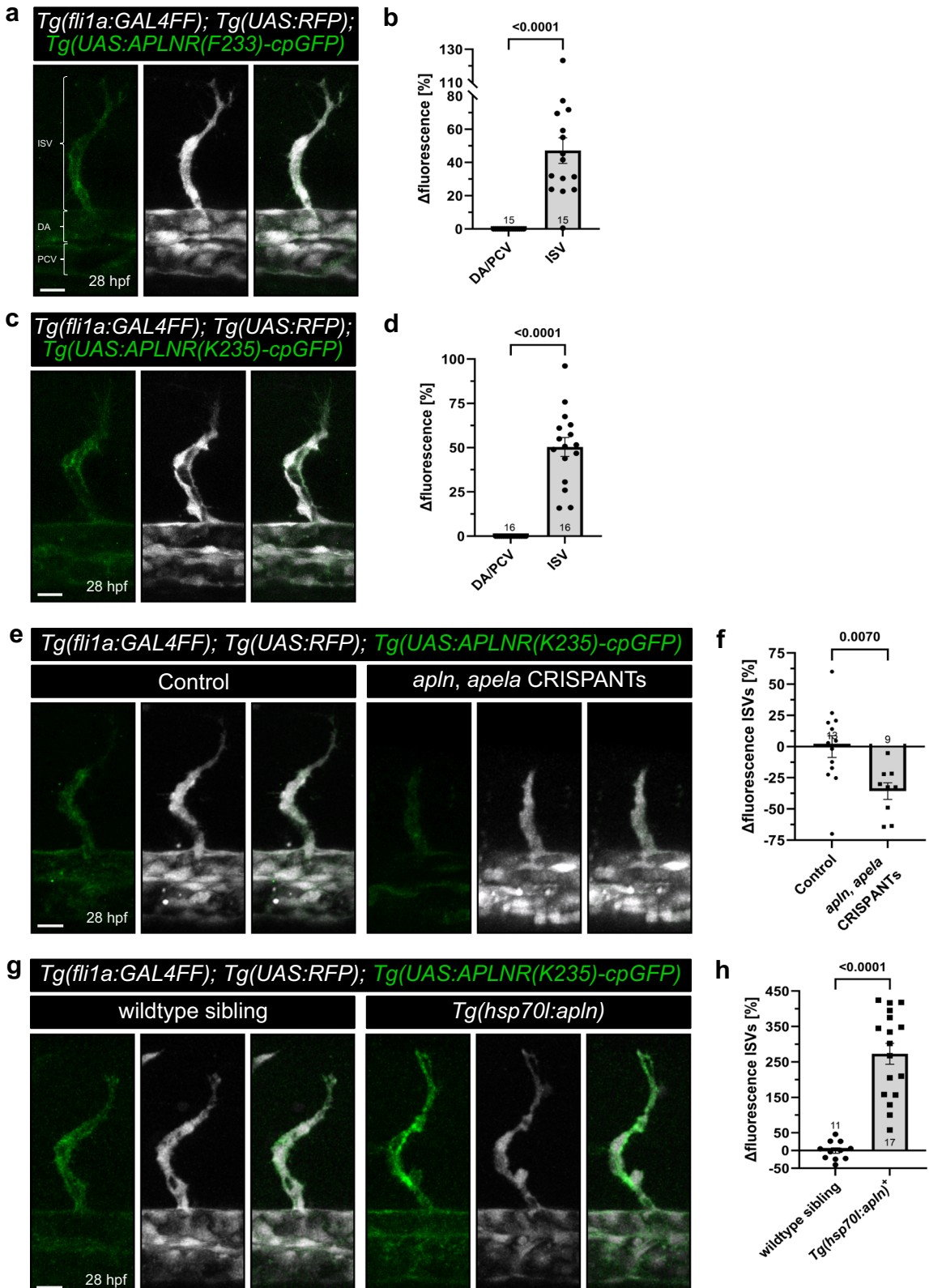

assessed the in vivo applicability of the ratiometric biosensors. We injected plasmids encoding either *UAS:*APLNR(K235)-cpGFP-mScarlet-I3 and *UAS:*APLNR(K235)-cpGFP-p2A-mScarlet-I3 into *Tg(fli1a:GAL4FF)* zebrafish embryos (Fig. 7e). This approach results in a mosaic expression of the ratiometric biosensor within the vasculature. We imaged the trunk vasculature (Fig. 7f) and quantified APLNR(K235)-cpGFP and mScarlet-I3 intensities in ISVs as well

as the DA and PCV at 28 hpf (Fig. 7g, h). Consistent with our observations in stable transgenic APLNR-cpGFP zebrafish lines, the ratiometric APLNR-cpGFP-mScarlet-I3 biosensors revealed higher cpGFP intensities in ISVs compared to the DA and PCV. Interestingly, the observed cpGFP intensity was more pronounced in the p2A variant with a 70% increase (Fig. 7h) compared to the variant without a p2A with only a 35% increase (Fig. 7g).

**Fig. 5 | APLNR-cpGFP biosensors visualize endogenous Aplnr activity in vivo.** **a** Representative confocal projection images of blood vessels in the trunk of triple transgenic *Tg(fli1a:GAL4FF); Tg(UAS:RFP); Tg(UAS:APLNR(F233)-cpGFP)* zebrafish embryos at 28 hours post-fertilization (hpf). **b** Quantification of APLNR(F233)-cpGFP delta fluorescence of ISVs compared to the DA/PCV. Each dot represents the mean of five analyzed ISVs per embryo or of the DA/PCV. **c** Representative confocal projection images of blood vessels in the trunk of triple transgenic *Tg(fli1a:GAL4FF); Tg(UAS:RFP); Tg(UAS:APLNR(K235)-cpGFP)* zebrafish embryos at 28 hpf. **d** Quantification of APLNR(K235)-cpGFP delta fluorescence intensity of ISVs compared to the DA/PCV. Each dot represents the mean of five analyzed ISVs per embryo or of the DA/PCV. **e** Representative confocal projection images of blood vessels in the trunk of triple transgenic *Tg(fli1a:GAL4FF); Tg(UAS:RFP); Tg(UAS:APLNR(K235)-cpGFP)* zebrafish embryos injected with *apln, apela* CRISPANTs at 28 hpf. **f** Quantification of APLNR(K235)-cpGFP delta fluorescence of ISVs in *apln,*

*apela* CRISPANTs compared to control siblings. Each dot represents the mean of up to five analyzed ISVs per embryo. **g** Representative confocal projection images of blood vessels in the trunk of *Tg(fli1a:GAL4FF); Tg(UAS:RFP); Tg(UAS:APLNR(K235)-cpGFP); Tg(hsp70l:apln)* zebrafish embryos at 28 hpf. **h** Quantification of APLNR(K235)-cpGFP delta fluorescence of ISVs in Apelin ligand overexpression *Tg(hsp70l:apln)* embryos compared to control wildtype siblings. Each dot represents the mean of five analyzed ISVs per embryo. Data are presented as mean values ± SEM. (*N* number of embryos, *n* either ISVs or DA/PCV: **b** DA/PCV *N/n*: 15/15, ISV *N/n*: 15/75; **d** DA/PCV *N/n*: 16/16, ISV *N/n*: 16/80; **f** Control: DA/PCV *N/n*: 13/13, ISV *N/n*: 13/65; *apln, apela* CRISPANTs: DA/PCV *N/n*: 9/9, ISV *N/n*: 9/38; **h** wildtype siblings: DA/PCV *N/n*: 11/11, ISV *N/n*: 11/55; *Tg(hsp70l:apln)*: DA/PCV *N/n*: 17/17, ISV *N/n*: 17/85). Statistical analysis was performed by using a two-tailed unpaired Student's *t*-test with Welch's correction. Scale bars 15 μm. cpGFP circularly permuted GFP, ISV intersegmental vessel, DA dorsal aorta, PCV posterior cardinal vein.

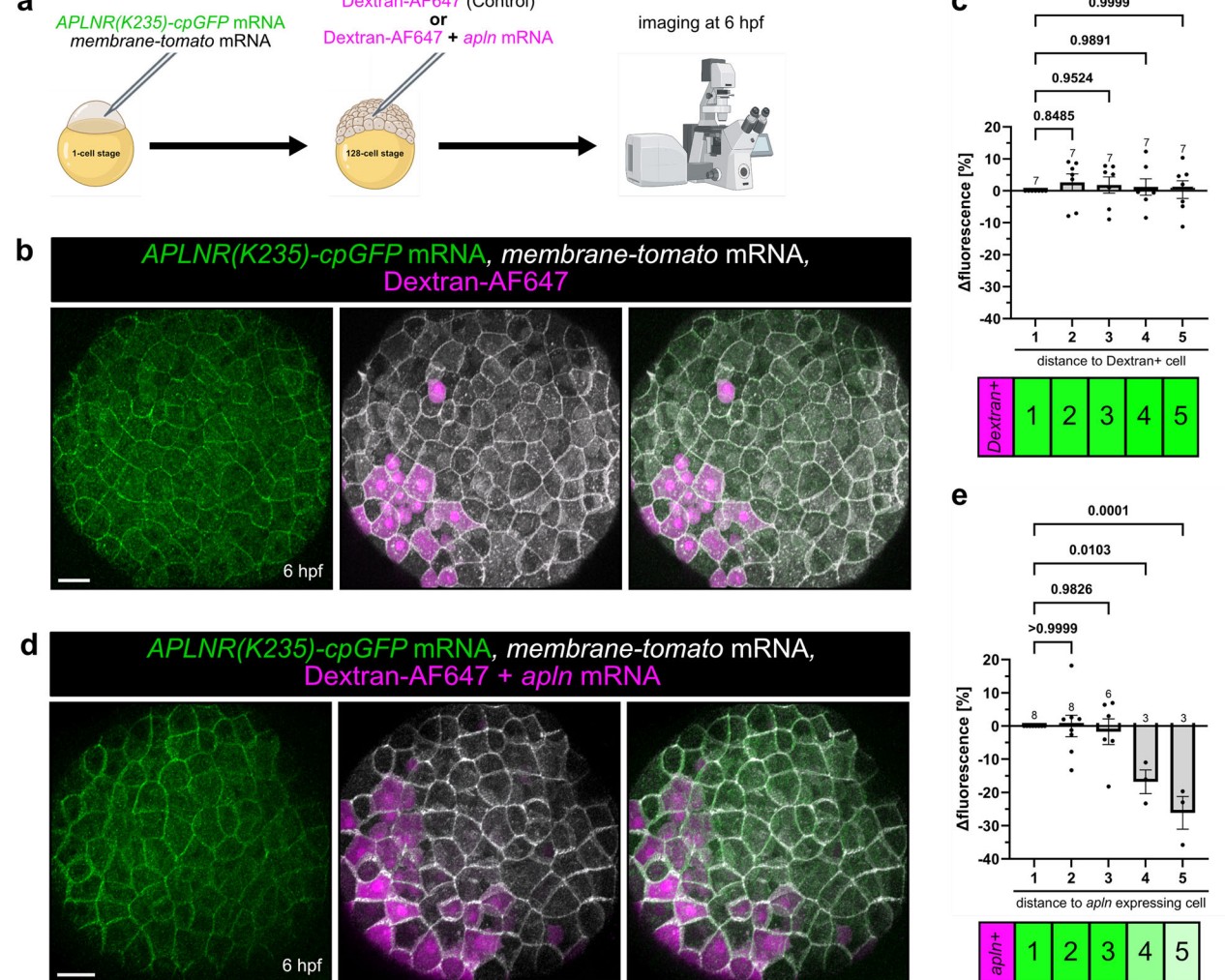

**Fig. 6 | Measuring an Apelin ligand gradient in vivo. a** Schematic illustration of the experiment. *APLNR(K235)-cpGFP* and *membrane-tomato* mRNA were injected into 1-cell stage zebrafish embryos. At the 128-cell stage, Dextran-AF647 or *apln* mRNA together with Dextran-AF647 were injected intracellularly in single blastomeres. Embryos were imaged at 6 hours post-fertilization (hpf). Representative confocal projection image of double-injected embryos with only Dextran-AF647 (**b**) or *apln* mRNA together with Dextran-AF647 (**d**) at 6 hpf. **c**, **e** Quantification of APLNR(K235)-cpGFP delta fluorescence intensity of single cells in relation to their distance to a Dextran positive cell (**c**) or an *apln*-expressing Dextran positive cell

(**e**). Each dot represents the mean of cells with the same distance within an embryo. A distance of 1 indicates a direct neighbor cell of a Dextran-AF647 positive cell. Data are presented as mean values ± SEM. (*N* number of embryos, *n* number of cells; **c** 1 - *N/n*: 7/59, 2 - *N/n*: 7/54, 3 - *N/n*: 7/52, 4 - *N/n*: 7/51, 5 - *N/n*: 7/51; **e** 1 - *N/n*: 8/64, 2 - *N/n*: 8/50, 3 - *N/n*: 6/31, 4 - *N/n*: 3/14, 5 - *N/n*: 3/10). Statistical analysis was performed by using ordinary One-way ANOVA, followed by Dunnett's multiple comparison correction. Scale bars 30 μm. cpGFP circularly permuted GFP, AF647 AlexaFluor647. Source data are provided as a Source Data file. **a** Created in BioRender. Schihada (2025) https://BioRender.com/hsvc4j3.

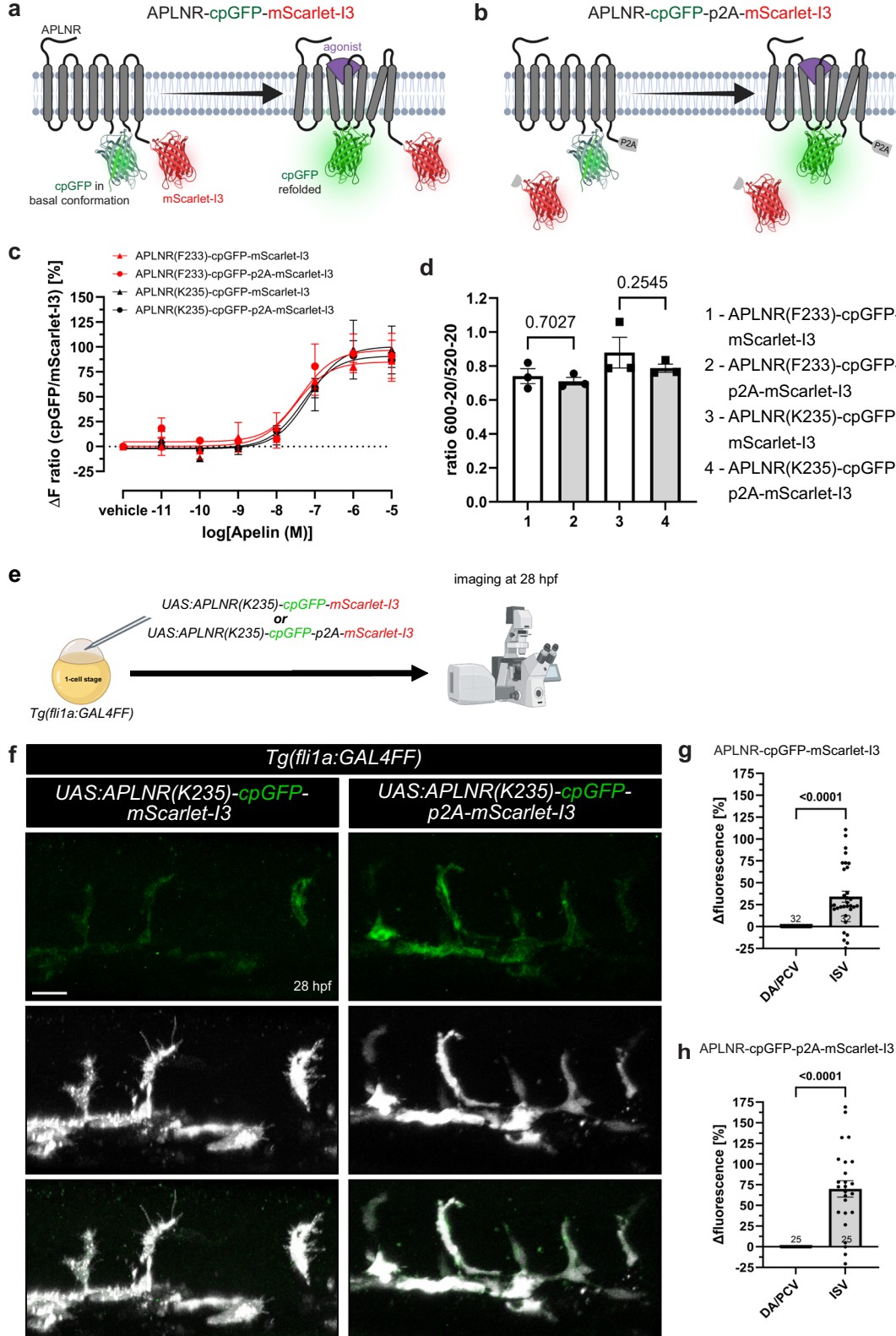

## Discussion

In this study, we developed and applied genetically encoded and in vivo-applicable conformational biosensors for the class A GPCR APLNR. Following the comparison of different biosensor variants, we focused on two APLNR-cpGFP biosensors due to their high sensitivity to Apelin and suitability for real-time imaging in living organisms. We pharmacologically characterized both APLNR-cpGFP biosensors in HEK293 cells and analyzed their performance in vivo using zebrafish embryos. In line with our in vitro results, our APLNR-cpGFP biosensors responded to Apelin and Apela and monitored endogenous Aplnr activity in zebrafish embryos. Moreover, we show that such biosensors can be employed to measure an Apelin gradient in vivo.

**Fig. 7 | Development and in vivo application of ratiometric APLNR-cpGFP-mScarlet-I3 biosensors. a, b** Schematic illustration of the ratiometric APLNR conformational biosensors design. mScarlet-I3 and p2A-mScarlet-I3 were integrated C-terminally into the APLNR-cpGFP biosensors, resulting in membrane-bound (**a**) and cytosolic (**b**) mScarlet-I3 localization. **c** HEK293A cells were transiently transfected with the respective ratiometric biosensor plasmid and stimulated with Apelin. Stimulation with Apelin led to a strong increase in cpGFP over mScarlet-I3 fluorescence intensity of each ratiometric biosensor variant. **d** Quantification of the FRET ratio of the ratiometric biosensor with and without a p2A self-cleaving peptide exhibited no significant differences, indicating that cpGFP and mScarlet-I3 are no FRET partner. **e** Schematic illustration of the procedure. Plasmids encoding either *UAS:APLNR(K23S)-cpGFP-mScarlet-I3 or UAS:APLNR(K23S)-cpGFP-p2A-mScarlet-I3* were injected into 1-cell stage *Tg(fli1a:-GAL4FF)* zebrafish embryos and imaged at 28 hours post fertilization (hpf).

**f, g** Quantification of APLNR(K235)-cpGFP over mScarlet-I3 fluorescence intensities in ISVs compared to the DA and PCV in *UAS:APLNR(K235)-cpGFP-mScarlet-I3* (**g**) or *UAS:APLNR(K235)-cpGFP-p2A-mScarlet-I3* (**h**) injected embryos at 28 hpf. Data in are presented as mean values ± SEM from three independent experiments conducted in transiently transfected HEK293A cells (**c, d**) or zebrafish embryos (*N* number of embryos, *n* either ISVs or DA/PCV: **g** DA/PCV *N/n*: 32/32, ISV *N/n*: 32/110; **h** DA/PCV *N/n*: 25/25, ISV *N/n*: 25/75). Statistical analysis was performed by using ordinary One-way ANOVA, followed Dunnett's multiple comparison correction (**d**) and two-tailed unpaired Student's *t*-test with Welch's correction (**g, h**). Scale bars 50 μm. ISV intersegmental vessel, DA dorsal aorta, PCV posterior cardinal vein, hpf hours post-fertilization, cpGFP circularly permuted GFP, HEK293 human embryonic kidney 293. Source data are provided as a Source Data file. **a, b** Created in BioRender. Schihada (2025) https://BioRender.com/odhtcw2. **e** Created in BioRender. Schihada (2025) https://BioRender.com/mz23zor.

In the first part of our study, we focused on the development and pharmacological validation of the APLNR biosensors. The APLNR-FRET (Fig. 1d) and -cpGFP (Fig. 1f) biosensors showed a robust response upon Apelin stimulation. These results show a favorable orientation of both energy partners in the FRET biosensor and the conformation-sensitive fluorescence intensity changes of cpGFP in response to APLNR activation. Surprisingly, none of the BRET biosensor variants showed sensitivity to Apelin treatment, although the insertion sites of the energy partners were identical to the ligand-sensitive APLNR-FRET constructs (Fig. 1e). We speculate that an unfavorable relative orientation of the BRET partners may be the underlying problem, and replacing mCitrine with other previously validated BRET acceptors (e.g., HaloTag NanoBRET™ 618[35,40]) may confirm this hypothesis. Although the APLNR-FRET biosensors successfully monitor APLNR activity, we further focused our analysis on the two APLNR-cpGFP biosensors (F233 and K235), due to their high sensitivity and superior signal-to-noise ratio. The cpGFP-based biosensors offer several advantages: (1) they provide an easy in vivo application, as they rely on the intensity changes of a single fluorescent protein, instead of the ratiometric measurement of two fluorophores as in the FRET biosensor, (2) the measurement can be performed by using a plate reader or standard imaging equipment like a confocal microscope and free-ware analysis software, like Fiji (ImageJ), (3) they are well-suited for high-throughput screenings to identify agonists and antagonists of the APLNR. Nevertheless, genetically encoded FRET-based biosensors are also in vivo-applicable and have the advantage of lifetime measurements using the fluorescence lifetime imaging microscopy (FLIM) technique. Therefore, both APLNR biosensor variants are suitable for use, but the biosensor choice needs to be considered depending on the performed assay.

Another interesting observation was that Apela and Apelin showed different potencies and efficacies in inducing conformational changes of the APLNR-cpGFP biosensors (Fig. 2c, d). While Apela activated APLNR biosensors with increased potency, the maximum fluorescence change was significantly lower than the response obtained with Apelin. This finding suggests that Apela and Apelin stabilize different receptor conformations, leading to potentially unique signaling profiles. In line with our results, a previous pharmacological study with various Apelin and Apela derivatives showed that these ligands induce distinct signaling profiles downstream of the APLNR[41]. Upon activation, GPCRs transmit signals intracellularly via coupling to G-proteins and recruiting Arrestins. In line with this, both APLNR-cpGFP biosensors showed $G_{i1}$ activation and Arrestin3 recruitment, although with reduced efficiency compared to the wildtype APLNR (Fig. 3). These reduced coupling efficiencies are likely due to the integration of the cpGFP module into the ICL3 of the APLNR, a domain known to be critical for transducer coupling[42–45]. Since our conformational biosensors function independently of the activation of specific downstream signaling

pathways, such as those involving G-proteins or cAMP, they are well-suited to investigate GPCRs with biased signaling properties or receptors that signal independently of G-proteins, such as Arrestins or other effectors. Notably, in cardiomyocytes, the Aplnr can be activated by mechanical stretch through a G-protein independent mechanism[46]. Therefore, our APLNR conformational biosensors are also well-suitable for investigating such non-canonical receptor signaling pathways.

The second part of our study explored the in vivo application of our APLNR biosensors. In zebrafish embryos, the APLNR(K235)-cpGFP biosensor responds to both endogenous ligands (Fig. 4) and enables real-time monitoring of endogenous Aplnr signaling (Fig. 5). Our biosensors provide precise temporal resolution of Aplnr signaling, facilitating the tracking of receptor dynamics in living organisms or model systems. This high-resolution monitoring offers insights into the molecular mechanisms underlying these cellular processes. During embryonic development, chemoattractant gradients guide cellular processes and pattern formation. These gradients can be established either through diffusion of the chemoattractant from the producing cells or via neighboring cells expressing scavenger receptors. Self-generated gradients have been implicated in various developmental contexts. A self-generated Apela gradient has been predicted through computational simulations and analysis of Aplnrb-sfGFP fusion protein internalization, suggesting its role in guiding mesodermal cell migration[3]. Similarly, a self-generated Cxcl12a gradient was identified through the turnover analysis of a fluorescent tandem Cxcr4b reporter[47,48]. These fluorescent tandem reporters visualize protein turnover, providing a strong indication of receptor activity. However, effective tools to directly visualize such gradients remain unavailable. To address this limitation, we employed our APLNR(K235)-cpGFP biosensor to monitor an Apelin gradient within living zebrafish embryos. Our analysis revealed a distance-dependent change in the fluorescence intensity of APLNR(K235)-cpGFP biosensor relative to the Apelin source (Fig. 6). In addition, we generated and validated advanced ratiometric APLNR biosensor versions for future studies (Fig. 7). These findings highlight the high efficiency and potency of our genetically encoded APLNR-cpGFP biosensor in visualizing a protein gradient in vivo. Monitoring ligand gradients enhances our understanding of how ligand gradients influence tissues and organ morphogenesis. Furthermore, the biosensors hold significant medical potential for advancing and refining targeted therapeutic strategies.

In summary, we successfully developed functional, genetically encoded APLNR conformation biosensors suitable for in vitro and in vivo applications. These biosensors expand the toolbox for investigating Apelin signaling in health and disease and hold significant potential for identifying and characterizing drugs targeting the APLNR.

# Methods

## Ethical statement

Zebrafish husbandry and maintenance was performed under standard conditions in accordance to institutional (Philipps-University Marburg) as well as national ethical and animal welfare guidelines approved by the ethics commission for animal experiments at the Regierungspräsidium Gießen, Germany, and the Federation of European Laboratory Animal Science Associations (FELASA) guidelines[49] (Akz.: V54-19 c 20 15 f 0 2 FB Biologie; V54-19 c 20 15 h 02 MR 17/1 Nr. A 1/2020; V54-19 c 20 15 h 01 MR 17/1 Nr. V 8/2022; V54-19 c 20 15 h 01 MR 17/1 Nr. G 102/2019).

## Agonists and antagonists

[Pyr$^1$]-Apelin-13 (BACHEM, 4029110), Apela(19-32) (MedChemExpress, HY-P2106A), ML-221 (Cayman chemical, Cay27313).

## Generation of APLNR-FRET, -BRET, and -cpGFP conformation biosensors

The human APLNR coding sequence (CDS) was amplified by PCR using the Takara PrimeStar polymerase (Takara, R045) and subsequently ligated into *pcDNA3.1* using the ClonExpress Ultra One Step Cloning Kit (Vazyme, C116). Additionally, the neomycin resistance cassette in the *pcDNA3.1* was replaced by a puromycin resistance cassette. We generated for each assay 4 different plasmids by inserting mCitrine (FRET and BRET) or cpGFP after each amino acid (F233, R234, K235, and E236) in the intracellular loop 3 of the APLNR. For the FRET and BRET biosensors, the APLNR CDS was C-terminally truncated after amino acid S343, while the full-length APLNR CDS was used for the cpGFP-based biosensors. Moreover, the cpGFP contains a C- and N-terminal linker sequence as previously described (LSSLI-cpGFP-NHDQL[30]). For the generation of the HA-tagged APLNR-cpGFP biosensors, the HA-Tag was integrated N-terminally downstream of the start codon into the APLNR(F233)-cpGFP and APLNR(K235)-cpGFP biosensors. For the generation of the ratiometric APLNR biosensors, mScarlet-I3 and p2A-mScarlet-I3 were integrated C-terminally upstream of the stop codon into the APLNR(F233)-cpGFP and APLNR(K235)-cpGFP biosensors. The primers used for cloning are listed in Supplementary Table 1.

## HEK293 cell culture and transient transfection

HEK293A (Cytion, 305070) and HEK293T (kind gift from Martin Lohse (Würzburg)) cells were cultured in DMEM (Gibco, 31966047) containing 10% FBS and 100 µg/ml penicillin-streptomycin mix (Gibco, 15140-122) and incubated at 37 °C with 5% $CO_2$. Cells were transiently transfected in suspension by mixing each ml of resuspended cells (300,000 cells/ml) with a mixture of 1 µg total DNA and 3 µl polyethyleneimine (PEI) stock solution (1 mg/ml in water) in 100 µl Opti-MEM and directly seeded into microtiter wells. Cells were then cultured for 48 h in complemented DMEM at 37 °C with 5% $CO_2$ until further proceeding.

## Generation of stable APLNR-cpGFP HEK293T cell lines

Stable HEK293T cell lines expressing APLNR-cpGFP were generated by plasmid transfection of either *CMV:APLNR(F233)-cpGFP* or *CMV:APLNR(K235)-cpGFP* using Lipofectamine2000 (Invitrogen, 11668019). HEK293T cells were cultured for five days in supplemented DMEM medium without antibiotics. After five days post-transfection, the medium was supplemented with 2 µg/ml puromycin for cell selection. After two weeks of selection, puromycin concentration was reduced to 1 µg/ml, which was also the final concentration for culturing the stable HEK293T cell lines.

## Single-cell FRET measurements

HEK293T cells were transfected 2 days before the measurement with plasmids encoding the respective APLNR-FRET biosensor under the control of the CMV promoter. Cells were harvested using trypsin and seeded on round poly-l-lysine (PLL)-coated glass coverslips the day before the measurement. Right before the measurements, the cultivation medium was discarded, and the cells were held in external buffer (137 mM NaCl, 5.4 mM KCl, 2 mM $CaCl_2$, 1 mM $MgCl_2$, 10 mM HEPES, pH 7.3). Single-cell FRET measurements were performed at a previously described microscope set-up[50] at a sampling frequency of 1 Hz. The cells were continuously superfused with external buffer with or without agonist using a pressurized perfusion system. For FRET measurements, samples were excited using an LED light source at 425 nm, and the emission was detected simultaneously at 485 (donor emission) and 535 nm (acceptor emission). The emission intensities were corrected for background fluorescence, bleed-through, and false excitation. The corrected acceptor emission was then divided by the corrected donor emission (referred to as "emission ratio"). The emission ratio was then plotted over time. To generate intensity color-coded images, the images were transferred in Fiji, converted to an 8-bit image, and the Lookup Table (LUT) was set to "16 colors".

## APLNR-mCitrine/NanoLuc BRET measurements

Cells transiently transfected with the APLNR-mCitrine/NanoLuc constructs were grown for 48 h in white 96-well plates and washed with 100 µl of HBSS. After washing, cells were incubated for 2 min with a 1/1000 solution of the furimazine stock solution (Promega; cat. no. N1662). Three baseline BRET reads were recorded and averaged prior to the addition of 10 µM [Pyr$^1$]-Apelin-13 or vehicle control and subsequent BRET reads. All experiments were conducted at 37 °C. BRET donor emission was quantified with a 470/80 nm filter. BRET acceptor emission was quantified using a 530/30 nm filter.

## Assessment of receptor surface expression through live-cell ELISA

To quantify cell surface receptor expression, HEK293 cells transfected with Gi1-CASE and pcDNA or N-terminally HA-tagged APLNR WT or APLNR-cpGFP constructs were grown for 48 h in transparent 96-well plates and washed once with 0.5% BSA in PBS. Next, cells were incubated with a polyclonal rabbit-anti-HA antibody (Proteintech, #51064-2-AP; diluted 1:1000) in 1% BSA–PBS for 1 h at 4 °C. Following incubation, the cells were washed three times with 0.5% BSA–PBS and incubated with a horseradish peroxidase-conjugated polyclonal swine anti-rabbit-HRP antibody (Dako, P0399; Lot #20058714; diluted 1:2000) in 1% BSA–PBS for 1 h at 4 °C. The cells were washed three times with 0.5% BSA/PBS, and 50 µl of the 3, 3′, 5, 5′ tetramethyl benzidine (TMB) substrate (BioLegend, San Diego, CA, USA) was added. Subsequently, the cells were incubated for 30 min and 50 µl of 2 M HCl was added. The absorbance was read at 450 nm using a BMG ClarioStar Plus plate reader.

## G$_{i1}$ heterotrimer dissociation BRET measurements

Cells transiently co-transfected with the G$_{i1}$ biosensor, Gi1-CASE[39], and wildtype APLNR or the respective APLNR-cpGFP biosensor with the indicated DNA amounts (50% Gi1-CASE combined with either 10 or 50% wildtype APLNR or 50% APLNR-cpGFP biosensors; pcDNA was used to add up 100% if required) were grown for 48 h in white 96-well plates and washed with 100 µl of HBSS. After washing, cells were incubated for 2 min with a 1/1000 solution of the furimazine stock solution (Promega; cat. no. N1662). Three baseline BRET reads were recorded and averaged prior to the addition of serial dilutions of [Pyr$^1$]-Apelin-13 or vehicle control and subsequent BRET reads. All experiments were conducted at 37 °C. BRET donor emission was quantified with a 470/80 nm filter. BRET acceptor emission was quantified using a 530/30 nm filter.

## APLNR-cpGFP measurements

Cells expressing the APLNR-cpGFP constructs were grown for 24 h (stable cells) or 48 h (transiently transfected cells) in black 96-well

plates and washed with 100 µl of HBSS. Three or fifteen baseline fluorescence reads were recorded and averaged prior to the manual addition or automated injection of ligands or vehicle control and subsequent fluorescent reads. All experiments were conducted at 37 °C. Fluorescence emission was quantified using 482/16 nm excitation and 520/10 nm emission filters. For experiments with the ratiometric biosensor versions, APLNR-cpGFP-mScarlet-I3, cpGFP, and mScarlet-I3 were alternately directly excited at 470/15 nm or 555/30 nm to quantify their emission intensities at 515/20 nm and 620/50 nm, respectively.

## Arrestin recruitment assays

APLNR-mediated Arrestin3 trafficking to the membrane was measured with a bystander BRET assay. For this purpose, SYFP2-CAAX was cloned into pcDNA3.1 by amplifying the SYFP2 coding sequence by PCR, digesting the PCR product with HindIII and NotI, and subsequently ligating the digested PCR product into the HindIII and NotI-digested pcDNA3.1 vector. The reverse primer contained the C-terminal CAAX motif. HEK293T cells were transiently transfected using PEI (1 mg/ml in water, pH 7.0). Just before the transfection, PEI was mixed with the constructs encoding bovine Arrestin3 C-terminally tagged with Nluc[51], SYFP2 fused to the N-terminus of the CAAX motif of the human Ras GTPase and either the APLNR wildtype, APLNR(F233)-cpGFP or APLNR(K235)-cpGFP in a ratio of 3:1. The resulting solutions for transfection were mixed with HEK293T cells resuspended in DMEM supplemented with 10% FCS, 2 mM L-glutamine, 100 U/ml penicillin and 0.1 mg/ml streptomycin (all from Capricorn Scientific GmbH, Ebsdorfergrund, Germany). 30,000 cells per well were seeded into a PLL-coated 96-well plate. After 48 h, the medium was removed and every well was washed once with HEPES buffer (137 mM NaCl, 5.4 mM KCl, 2 mM CaCl$_2$, 1 mM MgCl$_2$, 10 mM HEPES, pH 7.3). After that, 80 µl 6H-F-Coelenterazine diluted in HEPES buffer was added to the cells with a final concentration of 1 µM. The plate was immediately placed in a 37 °C pre-heated Tecan Spark 20 M plate reader (Tecan Group AG, Männedorf, Switzerland) and basal BRET was measured every 45 s for four times. Then 20 µl of either vehicle (HEPES buffer) or [Pyr$^1$]-Apelin-13 in HEPES buffer was added to final concentrations between 0.1 nM and 1 µM, followed by the measurement of 40 cycles with 45 s each. The fluorescence of the SYFP2 acceptor was measured between 520 nm and 590 nm, and the luminescence of the Nluc donor between 415 nm and 470 nm. The BRET values were calculated as the measured fluorescence of the acceptor divided by the luminescence of the donor and presented as the change in % over baseline. All values were normalized by subtraction of the BRET of the unstimulated cells from the stimulated cells. For creating the kinetic curves, the mean of the BRET values of four independent measurements was calculated. The concentration response curves were calculated in the same way and fitted using the equation log(agonist) vs. response (three parameters). 6H-F-coelenterazine was a kind gift of Prof. Dr. Wibke Diederich (Institute for Pharmaceutical Chemistry, University of Marburg, Germany).

mCherry-tagged bovine Arrestin3 was constructed by inserting XbaI and NotI sites directly before the ORF of mCherry and after the stop codon, respectively, using PCR. The PCR product was cut with XbaI and NotI and ligated to an Arrestin3-CFP construct[52] from which the CFP had been removed using the same restriction enzymes.

HEK293T cells seeded into 6 cm dishes were transiently transfected with 1 µg Arrestin3-mCherry and 1 µg of either the APLNR wildtype, APLNR(F233)-cpGFP, or APLNR(K235)-cpGFP using Effectene (Qiagen, Hilden, Germany). On the next day, the transfected cells were seeded on PLL-coated coverslips. Cells were imaged 2 days post-transfection using the 488 nm line of an argon laser (for GFP) and the 543 nm line of a diode laser (for mCherry) on a Leica SP5 confocal microscope equipped with a 63×/1.4 oil immersion lens. To assess Arrestin binding to the various APLNR constructs, cells were stimulated for 10 min with 1 µM [Pyr$^1$]-Apelin-13 before or during imaging.

## Zebrafish husbandry

Embryo maintenance took place under standard conditions at 28.5 °C in 1× E3 media (5 mM NaCl, 0.17 mM KCl, 0.33 mM CaCl$_2$, 0.33 mM MgSO$_4$, pH 7.2), and embryos were staged by hours post-fertilization (hpf)[53]. The following strains were used in this study[2,5,54,55]: AB wildtype; Tg(fli1a:GAL4FF)$^{ubs4}$; Tg(5xUAS:RFP)$^{nkuasrfp1a}$; Tg(hsp70l:apln)$^{mu269}$; Tg(hsp70l:apela, myl7:EGFP)$^{a143}$; Tg(5xUAS:APLNR(F233)-cpGFP)$^{mr34}$ (this study); Tg(5xUAS:APLNR(K235)-cpGFP)$^{mr35}$ (this study).

## Generation of transgenic Tg(5xUAS:APLNR(F233)-cpGFP)$^{mr34}$ and Tg(5xUAS:APLNR(K235)-cpGFP)$^{mr35}$ fish lines

For the generation of APLNR-cpGFP transgenic lines, inserts with homology arm overhangs were amplified by PCR using the Takara PrimeSTAR polymerase and subsequently ligated into the pminiTol2 entry vector using the ClonExpress Ultra One Step Cloning Kit. 40 pg of DNA of the plasmid and 30 pg of tol2 mRNA were co-injected into 1-cell stage zebrafish embryos for stable germline transmission, respectively. The primers used for cloning are listed in Supplementary Table 2.

## mRNA synthesis and microinjections

APLNR-cpGFP and APLNR-EGFP CDS were amplified by PCR using the Takara PrimeStar polymerase and ligated into pCS2+ using the ClonExpress Ultra One Step Cloning Kit. Primers used for cloning are listed in Supplementary Table 2. mRNA was synthesized using the mMES-SAGE mMACHINE SP6 transcription kit (Invitrogen, AM1340) and 200 pg/embryo membrane-tomato mRNA and 600 pg/embryo APLNR(K235)-cpGFP or APLNR-EGFP mRNA were injected into 1-cell-stage embryos in a volume of 4 nL. 200 pg apln mRNA and 1 ng 10 kDa Dextran-AlexaFluor647 (Invitrogen; D22914) were injected in single blastomeres in a volume of 0.5 nL at the 128-cell stage. For the ratiometric biosensors, 40 pg of the respective biosensor plasmid DNA was intracellularly co-injected with 30 pg tol2 mRNA into 1-cell-stage zebrafish embryos in a volume of 2 nL.

## crRNA design

To generate double knockouts for apln and apela, three synthetic CRISPR RNAs (crRNA, see Supplementary Table 4) targeting exons 1 and 2 of each gene were annealed to trans-activating RNA (trRNA) and intracellularly co-injected with Cas9 protein (5 µM) into 1-cell-stage zebrafish embryos[56]. For apln 2.5 fmol and for apela 0.625 fmol of the crRNA-trRNA complexes were injected per embryo. The Cas9 protein without crRNA-trRNA was used as a control. Cas9 protein and crRNA-trRNA were annealed at 37 °C for 10 min before injection. Functionality of the crRNAs was tested before the experiment by phenotypic screening. The pre-designed synthetic crRNAs and the Cas9 protein were purchased from IDT. The crRNAs were selected based on the highest on/off-target score via the IDT web tool.

## Tg(hsp70l:apln) or Tg(hsp70l:apela) overexpression

In order to temporally and ubiquitously overexpress the Apelin or Apela ligand, transgenic Tg(hsp70l:apln) or Tg(hsp70l:apela, myl7:EGFP) zebrafish were outcrossed with AB wildtype fish. To induce the ligand overexpression, offspring embryos were subjected to a single heat shock at 37 °C for 30 min at 5.5 hpf or 45 min at 27 hpf and subsequently imaged under a confocal microscope.

## Zebrafish microscopy

Zebrafish embryos were mounted in 1% low-melt agarose in glass-bottom dishes. 1× E3 media and agarose were supplemented with 19.2 mg/L Tricaine (Sigma Aldrich, E10521) for anesthesia. Embryos imaged at 28 hpf were treated with 0.1% (w/v) N-phenylthiourea (Sigma Aldrich, P7629) from 24 hpf onward to prevent pigmentation. Fluorescent confocal images were acquired on a Leica Stellaris 8 confocal microscope equipped with an HC FLUOTAR L VISIR 25×/0.95 WATER

objective. Maximum projection images were obtained using the Imaris Viewer 9.7.2 software (Bitplane). Fluorescent and brightfield images of the whole larvae and trunk were acquired using the Nikon SMZ18 stereo microscope equipped with a SHR Plan Apo 1× WD:60 objective and a DS Qi2 camera.

## APLNR-cpGFP biosensor quantification in vivo

The mean gray value of cpGFP and RFP or membrane-tomato in a region of interest was measured using Fiji (relative intensity), the background was subtracted (absolute intensity), respectively, and the absolute cpGFP fluorescence intensities were normalized to the absolute RFP or membrane-tomato fluorescence intensities. For transgenic cpGFP fluorescence intensity quantification in the blood vessels of the zebrafish trunk, five ISVs, as well as the DA and PCV, were measured per embryo. For injections, up to five ISVs were quantified per embryo. The mean value of ISVs' cpGFP fluorescence intensities was normalized to the fluorescence intensities of the DA/PCV. To enable a direct comparison between experimental groups in the CRISPANTs and Apelin ligand overexpression experiment at 28 hpf, we normalized the cpGFP fluorescence intensities of each ISV in *apln, apela* CRISPANTs, and *Tg(hsp70l:apln)* embryos to the mean cpGFP fluorescence intensity of control siblings' ISVs. For the ubiquitous ligand overexpression experiments at 6 hpf, 10 single cells per embryo were measured, and the mean was calculated for each embryo. Wildtype sibling cpGFP fluorescence intensities were used for normalization to quantify delta cpGFP fluorescence of *Tg(hsp70l:apln)* and *Tg(hsp70l:apela)* transgenic embryos, respectively. For the Apelin gradient experiment, a variable number of single cells per embryo were measured. Within each embryo, the mean was calculated for cells with the same distance. The cpGFP fluorescence intensity of a direct neighbor cell (distance =1) of an Apelin-Dextran-AlexaFluor647-positive cell was used for normalization to quantify the delta cpGFP fluorescence of cells with a greater distance (distance ≥2).

## Genotyping

For genotyping, single embryos were transferred into 30 µl (before 24 hpf) or 80 µl (after 24 hpf) 50 mM NaOH and boiled for 10 min at 95 °C to extract genomic DNA. Subsequently, 10% (v/v) of 1 M Tris-HCl (pH 8.0) was added to neutralize the pH. Genotyping for the transgenes *Tg(hsp70l:apln)*[5] or *Tg(hsp70l:apela)* was done by PCR. Primers used for the genotyping are listed in Supplementary Table 3.

## Statistical analysis and data presentation

FRET and BRET ratios were defined as acceptor emission/donor emission. To quantify ligand-induced changes, ΔFRET, ΔBRET, and Δfluorescence were calculated for each well as a percent over basal $(([(Ratio_{stim} - Ratio_{basal})/Ratio_{basal}] \times 100)$ for FRET and BRET; $([(fluorescence_{stim} - fluorescence_{basal})/fluorescence_{basal}] \times 100)$ for fluorescence). Subsequently, the average ΔFRET/ΔBRET/Δfluorescence of vehicle control was subtracted. Statistical analysis was performed using the Prism10 (GraphPad) software. Sample mean values were compared to normalized control values. Two-tailed unpaired Student's *t*-test with Welch's correction was used to compare two means. Ordinary One-way ANOVA, followed by Dunnett's multiple comparison correction, or 2-way ANOVA followed by Šidák multiple comparison correction, was used to compare multiple conditions. Data are shown as mean ± standard error of mean (SEM) or standard deviation (StD) as indicated in the respective figures. No data were excluded, and no statistical method was used to predetermine sample size. Experimental groups were not pre-selected based on the genotype or phenotype and were thus randomized. Investigators were blinded to group allocation during data collection as well as quantification.

## Reporting summary

Further information on research design is available in the Nature Portfolio Reporting Summary linked to this article.

## Data availability

All data are available in the Supplementary Information and Source data file. Source data are provided with this paper.

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

## Acknowledgements

We would like to thank S. Baumeister, F. Lehne, J. Malchow, N. Groos, and J. Ravindra for feedback, and S. Fischer, S. Engel, and R. Löchel for technical support. Funding: Microscopy was performed with the support of the Centre for Advanced Light Microscopy (CALM) Marburg [funded by the DFG (German Research Foundation); 446988475]. Open Access funding was provided by the Open Access Publishing Fund of Philipps-University Marburg.

## Author contributions

Conceptualization: L.H., H.S., and C.S.M.H.; Methodology: L.H., H.S., M.K., C.K., S.E., and C.S.M.H.; Investigation: L.H., H.S., M.K., S.E., J.E., C.K., and C.S.M.H.; Formal analysis: L.H., H.S., M.K., S.E., and C.K.; Resources: C.S.M.H.; Writing—original draft: L.H. and H.S.; Writing—review and editing: L.H., H.S., J.E., C.K., P.K., M.B., and C.S.M.H.; Manuscript approval: all authors.; Supervision: C.S.M.H. Funding acquisition: C.S.M.H.

## Funding

## Competing interests

The authors declare no competing interests.
