## [Transparent Peer Review file · Nature Communications]

In vivo measurement of an Apelin gradient with a genetically encoded APLNR conformation biosensor

Corresponding Author: Professor Christian Helker

Version 0:

Reviewer comments:

Reviewer #1

(Remarks to the Author)

Herdt L, et al. demonstrate the development of APLNR (Apelin receptor) biosensor and its usefulness in detecting APLNR ligand gradient in vivo. The authors have previously investigated the important biological roles of APLNR in cardiovascular development. Thus, they tried to develop APLNR activation-monitoring probes that enabled them to estimate its ligand gradient in vivo. Among the APLNR conformation-sensitive biosensor probes, circularly permuted GFP (cpGFP) showed the best signal-to-noise ratio before and after ligand stimulation. Therefore, they showed the usefulness of the cpGFP-based APLNR probe to monitor the activation of APLNR in the intersomitic vessel growth and to estimate the gradient of APLNR ligands in vivo.

The most striking result is the establishment of APLNR activation-monitoring probe after developing FEET-, BRET-, and cp-based probes. Recent reports on G protein-coupled receptor (GPCR) activation monitoring probes prove the contribution of these probes to investigate the biological function of GPCR in vivo. Thus, the probes the authors developed in the present study will contribute to the research on physiological and pathological angiogenesis where APLNR is involved. There are a few points that might be considered and added to the present manuscript in order to make the present study more biologically reliable and to convince readers that endogenous ligand gradient can be precisely monitored using the APLNR-cpGFP in vivo. Given the title of this manuscript, the authors seem to emphasize the importance of ligand gradient for APLNR.

Major concerns and points that might be clarified

1. The species difference of APLNR and its ligands should be tested.

The synthetic ligands used in the experiments are for ligands of mammalian APLNR. The APLNR cDNA used for the APLNR activation biosensors is supposed to be derived from mouse or human. Thus, the results of cultured cells (HEK293T) overexpressing APLNR-cpGFP and stimulated with synthetic ligands are logically examined. However, Does the APLNR-cpGFP reflect the zebrafish APLNR activation? There are three APLNR in zebrafish; *aplnrA*, *aplnrB*, and *aplnr2*. In addition, does the APLNR-cpGFP used in the present study respond to the zebrafish Apelin and Apela? These should be examined in cultured cells to show the usefulness of this probe when used in living zebrafish embryos.

233 FRKE 236 amino acid sequence is not present in *zAplnrA*, *zAplnrB* and *zAplnr2*.

To show the endogenous gradient, the authors need to demonstrate the effect of depletion of ApIn and Apela (zebrafish APLNR ligands) on the change of fluorescence intensity of APLNR-cpGFP.

2. The authors have investigated the *Aplnr* in the common cardinal vein (CCV) formation. They can show how the APLNR-cpGFP activation is changed during endothelial cell migration when they form CCV.

3. delta fluorescence (Y-axis) shown in Fig. 2a is different from that shown in Fig. 2c. While the delta fluorescence when cells were stimulated with 10^{-6} M Apelin is about 100%, that when stimulated with 10^{-6} M, 10^{-5} M, and 10^{-4} M was about 150%. Why is this difference observed?

Minor points

i) The plasma membrane localization of Arrestin3-mCherry indicated by red arrowheads are clear in APLNRWT but not in APLNR (F233-cpGFP and -APLNR(K235)-cpGFP). If the authors have magnified views, they can show clearer membrane localization in Fig.3.

- ii) In the method section, species of cDNA should be described.
- iii) Do the authors consider that Apln translated from injected apln mRNA in the cells is secreted outside of the cells (Fig.6).
- iv) Endogenous Apln and Apela can be monitored by the knock-in of fluorescence in the 3' of exon that is cleaved from the precursor peptide. Can the authors develop the ligand monitoring line to compare the ligand biosensor probe-expressing lines with the APLNR-cpGFP in the present study?

Reviewer #2

(Remarks to the Author)

The manuscript by Herdt et al. entitled "In vivo measurement of an Apelin gradient with a genetically 1 encoded APLNR conformation biosensor" describes the development of a novel APLNR biosensor, which the authors applied in vivo in zebrafish embryos to visualize endogenous APLNR activity in growing blood vessels. This short study forms the basis for further studies, allowing to study Apelin signaling activity in health and disease. Before being considered for publication, the following points need to be addressed:

1. Fig. 3c: The expression level of Arrestin3-mCherry is always lower in the APLNR(F233)-cpGFP compared to the ANPLN(F235)-cpGFP cell line – has this been considered for the analysis?
2. Fig. 4: The change in fluorescence in the images looks much more pronounced in response to Apela compared to Apelin – however, this is not represented in the graph. Can the authors comment on that?
3. Fig. 5: the higher fluorescence in these experiments can be due to other factors, e.g., different biosensor environment like a different pH. To directly compare absolute fluorescence intensities, the biosensor needs to include another fluorophore that does not react to ligand binding for normalization. Thus, the authors should create a ratiometric biosensor based on the ANPLN(F235)-cpGFP biosensor and re-do the analysis in vivo.
4. Fig. 6: The same comment applies to the gradient measurements – how can the authors be sure that the expression level of the biosensor is equal across the embryo? Also here, the implementation of a ratiometric biosensor would be beneficial. Furthermore, the authors should show a control injection with only Dextran to evaluate the specificity of the response.
5. To be able to use the biosensor in vivo, the authors should generate a mutant that allows binding but does not evoke a downstream response. This would allow to measure ligand availability without overexpressing a functional GPCR.

Reviewer #3

(Remarks to the Author)

The Apelin receptor plays important roles in various developmental and pathological processes including cardiovascular development. In this manuscript, the author engineered the biosensor of Apelin signaling and argue that cpGFP-based fusion constructs, called APLNR-cpGFP biosensors, can monitor Apelin receptor (Aplnr) activity in cell lines and zebrafish. Since the downstream signaling of Aplnr is not well understood, such biosensors would be beneficial to advance Apelin-Aplnr research in broader fields. Overall the experiments are well designed. In particular, the analyses using cultured cells are elaborate, including the measurement of IC50. Although they show that these biosensors respond to externally expressed Apelin in zebrafish as well as in culture cells, it is still unclear whether they can respond to physiological levels of Aplnr activation in vivo. In order to publish in Nature Communications, I feel it is necessary to show clearly that these biosensors can monitor endogenous Apelin and/or Apela.

Major comments

- Current Figure 5 data alone do not sufficiently demonstrate that APLNR-cpGFP biosensors reflect endogenous Aplnr activity in vivo. The best way to find out is to see if the loss of function of Apln and/or Apela reduces signals in the ISVs. In addition, they should normalize APLNR-cpGFP signals, localized in the plasma membranes, not with RFP (appeared to be localized in nuclei and cytoplasm), but with a membrane-localized fluorescent protein as they did in HEK293 or zebrafish early embryos.
- The authors found ~20% increase by 10 uM Apelin using HEK293 in Figure 1f, and ~25% increase by heat shock-induced Apelin using zebrafish in Figure 4c. Thus, the increase was not pronounced, and there was a large variation in fluorescence intensities from about -50% to 160% as shown in Fig. 4c,d. Is it possible to extract physiological Aplnr activity from background GFP signals at cellular levels in vivo?

Minor comments

- In Figure 5a, c, they show only a single ISV. The authors had better indicate a broader area.
- In Figure 3, the authors compared the effects of APLNR WT and APLNR-cpGFP biosensors in cultured cells, but to discuss the difference, they had better show whether the protein levels are comparable, such as western blotting for an apelin antibody.

Version 1:

Reviewer comments:

Reviewer #1

(Remarks to the Author)

Herd, L. et al. have addressed the major concerns from the three reviewers. Both reviewers #1 and #3 were concerned with the responses of APLNR-circularly permuted GFP probes (APLNR-cpGFP) to endogenous APLNR ligands. To test whether APLNR-cpGFP can monitor the endogenous ligand gradient *in vivo*, they examined the effect of depletion of Apelin and Apela on the activity of APLNR-cpGFP. In Apelin/Apela CRISPANTS, the APLNR-cpGFP was decreased (new Figure 5e). In addition, they examined the effect of overexpression of APLNR ligands on APLNR-coGFP (new Figure 5g and h). These data indicate that APLNR-cpGFP fluorescence clearly reflects the responses of the probes to endogenous APLNR ligands. Moreover, to address the concern from reviewer #2 about the expression difference of the probes presumably under cell circumstances, they developed new ratiometric biosensors. These new data fortify the usefulness of APLNR conformation biosensors that they developed to monitor APLNR activity *in vivo*. Given the new results, the authors successfully addressed most of the concerns from the reviewers. For the future study, the authors are encouraged to knock-in the probe in the endogenous loci of zebrafish APLNR to avoid the effect of overexpression of probes using GAL4/UAS system.

The following points should be carefully described.

1. The authors still need to clarify the APLNR-cpGFP biosensor quantification *in vivo* described in the methods (from line 599). The normalization of fluorescence intensity using DA/PCV is unclear. It is also unclear how the authors additionally normalized the intensity of ISV to compare that of the control and with that of the CRISPANTS (Figure 5).
2. There is no explanation of arrowheads of Figure 3 in the legends.

Reviewer #2

(Remarks to the Author)

The authors have addressed all my comments and I am happy to advise publication of the manuscript.

Reviewer #3

(Remarks to the Author)

We appreciate the reviewer's supportive comments and practical suggestions to improve our manuscript.

Reviewer: 1

General comment

Herdt L, et al. demonstrate the development of APLNR (Apelin receptor) biosensor and its usefulness in detecting APLNR ligand gradient in vivo. The authors have previously investigated the important biological roles of APLNR in cardiovascular development. Thus, they tried to develop APLNR activation-monitoring probes that enabled them to estimate its ligand gradient in vivo. Among the APLNR conformation-sensitive biosensor probes, circularly permuted GFP (cpGFP) showed the best signal-to-noise ratio before and after ligand stimulation. Therefore, they showed the usefulness of the cpGFP-based APLNR probe to monitor the activation of APLNR in the intersomitic vessel growth and to estimate the gradient of APLNR ligands in vivo.

The most striking result is the establishment of APLNR activation-monitoring probe after developing FEET-, BRET-, and cp-based probes. Recent reports on G protein-coupled receptor (GPCR) activation monitoring probes prove the contribution of these probes to investigate the biological function of GPCR in vivo. Thus, the probes the authors developed in the present study will contribute to the research on physiological and pathological angiogenesis where APLNR is involved. There are a few points that might be considered and added to the present manuscript in order to make the present study more biologically reliable and to convince readers that endogenous ligand gradient can be precisely monitored using the APLNR-cpGFP in vivo. Given the title of this manuscript, the authors seem to emphasize the importance of ligand gradient for APLNR.

Major comments:

1. *The species difference of APLNR and its ligands should be tested.*

*The synthetic ligands used in the experiments are for ligands of mammalian APLNR. The APLNR cDNA used for the APLNR activation biosensors is supposed to be derived from mouse or human. Thus, the results of cultured cells (HEK293T) overexpressing APLNR-cpGFP and stimulated with synthetic ligands are logically examined. However, Does the APLNR-cpGFP reflect the zebrafish APLNR activation? There are three APLNR in zebrafish; *aplnra*, *aplnrb*, and *aplnr2*. In addition, does the APLNR-cpGFP used in the present study respond to the zebrafish Apelin and Apela? These should be examined in cultured cells to show the usefulness of this probe when used in living zebrafish embryos.*

233 FRKE 236 amino acid sequence is not present in zAplnra, zAplnrb and zAplnr2.

*To show the endogenous gradient, the authors need to demonstrate the effect of depletion of *Apln* and *Apela* (zebrafish APLNR ligands) on the change of fluorescence intensity of APLNR-cpGFP.*

We thank the Reviewer for the valid suggestions. The APLNR-cpGFP biosensors are based on the human coding sequence. In Fig. 4, we show that the APLNR-cpGFP biosensor respond to the stimulation of zebrafish Apelin and Apela via the transgenic heatshock inducible lines in zebrafish embryos. These zebrafish lines overexpress the zebrafish Apelin and Apela ligand.

Furthermore, we validated the APLNR(K235)-cpGFP biosensor in embryos depleted for the ligands *apln* and *apela* (knockout) and detected a decrease in the APLNR(K235)-cpGFP biosensor activity. We added this new result in Figure 5 e. In addition, we overexpressed the zebrafish Apelin ligand with the heat-inducible *Tg(hsp70l:apln)* line,

which led to an increase of the biosensor intensity in ISVs compared to control siblings. We added this new result in Figure 5 g, h. Together our new results underline the APLNR-cpGFP biosensors response to zebrafish Apelin and Apela and its activity modulation in absence and presence of the ligands.

2. The authors have investigated the *Aplnr* in the common cardinal vein (CCV) formation. They can show how the APLNR-cpGFP activation is changed during endothelial cell migration when they form CCV.

We thank the reviewer for this suggestion. The formation of the CCV is not dependent on Apelin signaling (Figure R1). Therefore, we did not investigate the APLNR-cpGFP biosensor in the CCV.

Figure R1. Apelin signaling is dispensable for CCV development. siblings N = 12 larvae; *apl n* -/ - N = 12 larvae. Scale bar: 50 μ m.

3. delta fluorescence (Y-axis) shown in Fig. 2a is different from that shown in Fig. 2c. While the delta fluorescence when cells were stimulated with 10-6M Apelin is about 100%, that when stimulated with 10-6M, 10-5M, and 10-4M was about 150%. Why is this difference observed?

We thank the reviewer for this legitimate point. The experiments in 2a and b vs. 2c and d have been performed with slightly adapted plate reader settings to account for the higher temporal resolution required for the time course data in 2a and b. Specifically, the number of flashes per data point (equivalent to the integration time in luminescence-based experiments) has been reduced for the data shown in 2a and b to enable more measurements within the same time frame. While we believe that this technical adaptation may have contributed to the altered signal amplitude, we cannot rule out the impact of other variables such as the passage of the stable sensor cell line.

Minor comments

i) The plasma membrane localization of Arrestin3-mCherry indicated by red arrowheads are clear in APLNRWT but not in APLNR (F233-cpGFP and -APLNR(K235)-cpGFP. If the authors have magnified views, they can show clearer membrane localization in Fig.3.

We thank the reviewer for their helpful suggestion. We added magnified areas of the cells to better visualize membrane-localized Arrestin3-mCherry after Apelin stimulation in Figure 3 c.

ii) In the method section, species of cDNA should be described.

We thank the reviewer for this comment. We added the species of the cDNA in the method section.

iii) Do the authors consider that Apln translated from injected apln mRNA in the cells is secreted outside of the cells (Fig.6).

We thank the reviewer for this legitimate suggestion. The mRNA contains the full coding sequence including the signal peptide sequence. Therefore, the Apelin protein is secreted after translation.

iv) Endogenous Apln and Apela can be monitored by the knock-in of fluorescence in the 3' of exon that is cleaved from the precursor peptide. Can the authors develop the ligand monitoring line to compare the ligand biosensor probe-expressing lines with the APLNR-cpGFP in the present study?

We thank the reviewer for this valid point. To date, the processing mechanism of Apelin is not fully elucidated, but processing of the Apelin prepropeptide take place intracellularly (Demoures et al. 2025; Kleinz and Davenport 2005; Shin, Kenward, and Rainey 2018). Therefore, an intracellular processing would result in the accumulation of the fluorescent protein in the cytosol of the *apl*n-expressing cells. Such a ligand monitoring line would be similar to our previously published transgenic reporters (Helker et al. 2020; Malchow et al. 2024). By using these transgenic reporters, we have shown that in the zebrafish trunk, Apelin is expressed from neural progenitors, while the *Aplnr* is expressed by the blood vessels, which is required for spinal cord vascularization (Helker et al. 2020; Malchow et al. 2024).

Reviewer: 2

General comment

The manuscript by Herdt et al. entitled "In vivo measurement of an Apelin gradient with a genetically encoded APLNR conformation biosensor" describes the development of a novel APLNR biosensor, which the authors applied in vivo in zebrafish embryos to visualize endogenous APLNR activity in growing blood vessels. This short study forms the basis for further studies, allowing to study Apelin signaling activity in health and disease. Before being considered for publication, the following points need to be addressed:

Major comments

1. Fig. 3c: The expression level of Arrestin3-mCherry is always lower in the APLNR(F233)-cpGFP compared to the ANPLN(F235)-cpGFP cell line – has this been considered for the analysis?

We thank the reviewer for this valid point. In order to assess the capacity of both APLNR biosensors to promote Arrestin3 translocation from the cytosol to the plasma membrane, we measured the relative fluorescence intensity at the plasma membrane vs. cytosol within one biological sample. Furthermore, the confocal images shown in Figure 3c are not used to provide a quantitative comparison of the signaling capacities of APLNR(F233)-cpGFP vs. APLNR(K235)-cpGFP. This is achieved through the BRET-based Arrestin3 recruitment data shown in Figure 3b, a ratiometric readout that, due to normalization to baseline conditions (i.e., before APLN treatment), internally controls for altered expression levels of the labeled Arrestin3.

2. Fig. 4: The change in fluorescence in the images looks much more pronounced in response to Apela compared to Apelin – however, this is not represented in the graph. Can the authors comment on that?

We thank the reviewer for this observation. We have adapted the Tg(hsp70l:apln) image in Figure 4 to better resolve individual cells and to more clearly display the expression of the APLNR(K235)-cpGFP biosensor. In the quantification graph, the overall fluorescence intensity changes are averaged over multiple embryos and regions of interest, which can dampen apparent differences visible in single images. We updated the image to a more representative one, better reflecting the typical fluorescence intensity changes observed across samples.

3. Fig. 5: the higher fluorescence in these experiments can be due to other factors, e.g., different biosensor environment like a different pH. To directly compare absolute fluorescence intensities, the biosensor needs to include another fluorophore that does not react to ligand binding for normalization. Thus, the authors should create a ratiometric biosensor based on the ANPLN(F235)-cpGFP biosensor and re-do the analysis *in vivo*.

We thank the reviewer for this legitimate suggestion. Since both the cpGFP fluorophore of the biosensor and the RFP are located within the cytosol, environmental factors such as local pH that could potentially interfere with the fluorescence signal are considered negligible. In addition, the RFP is expressed under the control of the UAS promotor as the biosensors and does not respond to ligand binding. Nevertheless, to rigorously exclude expression differences as a cause for the observed higher fluorescence intensities, we generated ratiometric APLNR-cpGFP-mScarlet-I3 biosensors. These were constructed either by direct C-terminal fusion to mScarlet-I3 for membrane localization or by using a self-cleaving p2A peptide to achieve cytosolic localization. We decided to generate both variants with and without the p2A linker, since mScarlet-I3 and cpGFP are potential FRET partners (Fig. 7a, b). We first validated the functionality and FRET behavior of the ratiometric biosensors in HEK293A cells. All four generated biosensor variants showed a comparable increase in the cpGFP/mScarlet-I3 fluorescence intensity ratio upon Apelin stimulation, with no detectable quenching of cpGFP fluorescence (Fig. 7c, d). Furthermore, we investigated the ratiometric APLNR(K235)-cpGFP-mScarlet-I3 biosensor variants *in vivo* and repeated the ISV experiments previously performed with the non-ratiometric stable zebrafish line. Both ratiometric biosensor versions (APLNR(K235)-cpGFP-mScarlet-I3 and APLNR(K235)-cpGFP-p2A-mScarlet-I3) confirmed our earlier results, demonstrating that cells in the ISVs indeed exhibit a higher APLNR-cpGFP biosensor activity compared to the cells in

the dorsal aorta (DA) and posterior cardinal vein (PCV). These new datasets have been included in the revised manuscript as a new Figure 7.

4. Fig. 6: The same comment applies to the gradient measurements – how can the authors be sure that the expression level of the biosensor is equal across the embryo? Also here, the implementation of a ratiometric biosensor would be beneficial. Furthermore, the authors should show a control injection with only Dextran to evaluate the specificity of the response.

We thank the reviewer for these valid suggestions. To show the uniform distribution of the injected biosensor mRNA, we generated an APLNR-EGFP construct that does not change GFP fluorescence intensity upon ligand binding. Stimulation with Apelin had no effect on APLNR-EGFP fluorescence intensity, indicating that the biosensor distribution is uniform across the embryo. We have now included this new data to the manuscript as a Supplementary Figure 4. Furthermore, since the ratiometric APLNR-cpGFP-mScarlet-I3 biosensors fully reproduced the relative fluorescence patterns and activity differences observed with the non-ratiometric biosensors (Fig. 7), we conclude that the original datasets are robust and not biased by expression variability. In addition, we performed control injections with Dextran alone. These experiments confirmed that in the absence of Apelin, we did not observe any distance-dependent changes in APLNR(K235)-cpGFP fluorescence intensity. These results further validate the specificity and sensitivity of the biosensor for detecting endogenous ligand gradients across cellular distances. The new control data have been added to the revised manuscript in Fig. 6b, c.

5. To be able to use the biosensor *in vivo*, the authors should generate a mutant that allows binding but does not evoke a downstream response. This would allow to measure ligand availability without overexpressing a functional GPCR.

We thank the reviewer for this suggestion. In our *in vivo* experiments, we did not observe any developmental or vascular abnormalities in biosensor-expressing embryos, suggesting no to minimal interference with endogenous signaling. In support of this, our *in vitro* characterization showed that APLNR-cpGFP biosensors induce substantially less Gi1 dissociation and β -Arrestin3 recruitment compared to the wild-type APLNR (Fig. 3). This indicates that cpGFP insertion partially uncouples the receptor from downstream signaling pathways, likely reducing biological activity *in vivo*. We also explored the generation of signaling-deficient receptor variants by mutating known phosphorylation sites. However, our mutagenesis of the phosphorylation sites (S335, S339, S345, S369) did not affect Arrestin3 recruitment (Fig. R2). To the best of our knowledge, a mutant APLNR version that is entirely uncoupled from both G protein- and Arrestin-dependent pathways—while still capable of ligand binding—remains undefined.

Fig. R2. Mutagenesis of APLNR phosphorylation sites has no impact on Arrestin3 recruitment. Arrestin3 recruitment BRET assay of APLNR wildtype (WT) and APLNR phosphorylation site mutants. No differences in Arrestin3 recruitment between the APLNR wildtype and APLNR mutant constructs were observed. Data are presented as mean values \pm StD from three independent experiments conducted in transiently transfected HEK293T cells.

Reviewer: 3

General comment

The Apelin receptor plays important roles in various developmental and pathological processes including cardiovascular development. In this manuscript, the author engineered the biosensor of Apelin signaling and argue that cpGFP-based fusion constructs, called APLNR-cpGFP biosensors, can monitor Apelin receptor (Aplnr) activity in cell lines and zebrafish. Since the downstream signaling of Aplnr is not well understood, such biosensors would be beneficial to advance Apelin-Aplnr research in broader fields. Overall the experiments are well designed. In particular, the analyses using cultured cells are elaborate, including the measurement of IC50. Although they show that these biosensors respond to externally expressed Apelin in zebrafish as well as in culture cells, it is still unclear whether they can respond to physiological levels of Aplnr activation in vivo. In order to publish in Nature Communications, I feel it is necessary to show clearly that these biosensors can monitor endogenous Apelin and/or Apela.

Major comments

Current Figure 5 data alone do not sufficiently demonstrate that APLNR-cpGFP biosensors reflect endogenous Aplnr activity in vivo. The best way to find out is to see if the loss of function of Apln and/or Apela reduces signals in the ISVs. In addition, they should normalize APLNR-cpGFP signals, localized in the plasma membranes, not with RFP (appeared to be localized in nuclei and cytoplasm), but with a membrane-localized fluorescent protein as they did in HEK293 or zebrafish early embryos.

We thank the reviewer for these valid suggestions. We validated the APLNR(K235)-cpGFP biosensor in embryos depleted for the ligands *apl*n and *apela* (knockout) and detected a decrease in the APLNR(K235)-cpGFP biosensor activity. We added this new result in Figure 5 e. In addition, we overexpressed the zebrafish Apelin ligand with the heat-inducible *Tg(hsp70l:apl*n) line, which led to an increase of the biosensor intensity in ISVs compared to control siblings. We added this new result in Figure 5 g, h. Together our new results underline the APLNR-cpGFP biosensors response to zebrafish Apelin and Apela and its activity modulation in absence and presence of the ligands.

Since both the cpGFP fluorophore of the biosensor and the RFP are located within the cytosol, environmental factors such as local pH that could potentially interfere with the fluorescence signal are considered negligible. In addition, the RFP is expressed under the control of the UAS promoter as the biosensors and does not respond to ligand binding. Nevertheless, to rigorously exclude expression differences as a cause for the observed higher fluorescence intensities, we generated ratiometric APLNR-cpGFP-mScarlet-I3 biosensors. These were constructed either by direct C-terminal fusion to mScarlet-I3 for membrane localization or by using a self-cleaving p2A peptide to achieve cytosolic localization. We decided to generate both variants with and without the p2A linker, since mScarlet-I3 and cpGFP are potential FRET partners (Fig. 7a, b). We first validated the functionality and FRET behavior of the ratiometric biosensors in HEK293A cells. All four generated biosensor variants showed a comparable increase in the cpGFP/mScarlet-I3 fluorescence intensity ratio upon Apelin stimulation, with no detectable quenching of cpGFP fluorescence (Fig. 7c, d). Furthermore, we investigated the ratiometric APLNR(K235)-cpGFP-mScarlet-I3 biosensor variants *in vivo* and repeated the ISV experiments previously performed with the non-ratiometric stable zebrafish line. Both ratiometric biosensor versions (APLNR(K235)-cpGFP-mScarlet-I3 and APLNR(K235)-cpGFP-p2A-mScarlet-I3) confirmed our earlier results, demonstrating that cells in the ISVs indeed exhibit a higher APLNR-cpGFP biosensor activity compared to the cells in the dorsal aorta (DA) and posterior cardinal vein (PCV). These new datasets have been included in the revised manuscript as a new Figure 7.

The authors found ~20% increase by 10 uM Apelin using HEK293 in Figure 1f, and ~25% increase by heat shock-induced Apelin using zebrafish in Figure 4c. Thus, the increase was not pronounced, and there was a large variation in fluorescence intensities from about -50% to 160% as shown in Fig. 4c,d. Is it possible to extract physiological Aplr activity from background GFP signals at cellular levels *in vivo*?

We thank the reviewer for this suggestion. Indeed, the fluorescence increases we observed in HEK293T cells (~20% in Fig. 1f) is moderate. However, it is important to note that the experiment shown in Fig. 1 was performed with transiently transfected HEK293 cells, whereas in Fig. 2 a we used a stable HEK293T cell line expressing the APLNR(K235)-cpGFP biosensor. In the stable cell line, we observed fluorescence increases of up to ~100% upon Apelin stimulation, indicating that the moderate fluorescence changes in Fig. 1 are likely caused by technical variability associated with transient transfection, rather than by fundamental limitations of the biosensor itself.

In the *in vivo* zebrafish setting, detecting cpGFP fluorescence changes at the single-cell level is challenging due to the optical and biological complexity of living tissues. However, we observed statistically significant increases in biosensor fluorescence following Apelin stimulation, showing that meaningful physiological Aplr activity can still be resolved *in vivo*. For our *in vivo* experiments, we did background subtraction of the cpGFP and normalized afterward cpGFP intensities to membrane-tomato intensities.

Minor comments

- In Figure 5a, c, they show only a single ISV. The authors had better indicate a broader area.

We thank the Reviewer for this comment. We imaged the vascular-specific APLNR-cpGFP biosensor expression within the whole embryo and in magnified images of the trunk. We added these new data to the manuscript in Supplementary Fig. 5.

In Figure 3, the authors compared the effects of APLNR WT and APLNR-cpGFP biosensors in cultured cells, but to discuss the difference, they had better show whether the protein levels are comparable, such as western blotting for an apelin antibody.

We thank the Reviewer for this helpful suggestion. We now generated HA-tagged APLNR-cpGFP biosensors and performed ELISA analysis to quantify and compare the surface expression levels of APLNR WT and the two APLNR-cpGFP biosensors. We could only detect a minor expression difference of the APLNR WT compared to the APLNR(F233)-cpGFP biosensor, but not to the APLNR(K235)-cpGFP biosensor. Moreover, we could not detect an expression difference between the F233 and K235 biosensor variants. Based on the ELISA expression results, we repeated the G-protein dissociation assay with adapted amounts of transfected plasmid DNA to account for the different surface expression levels and to provide data for a comparative analysis of the APLNR wildtype and the APLNR-cpGFP signaling capacities. Importantly, even after adjusting for surface expression, the results confirmed our previous findings regarding the reduced signaling efficiency of the APLNR-cpGFP biosensors compared to the wildtype receptor. These new datasets have been included in the revised manuscript as a new Supplementary Figure 2. We replaced the G-protein dissociation results in Fig. 3 a with the results of the new assay with adapted APLNR wildtype amounts.

References

- Demoures, Béatrice, Fabienne Soulet, Jean Descarpentrie, Isabel Galeano-Otero, José Sanchez Collado, Maria Casado, Tarik Smani, Alvaro González, Isabel Alves, Fabrice Lalloué, Bernard Masri, Estelle Rascol, Jean William Dupuy, Cyril Dourthe, Frédéric Saltel, Anne Aurélie Raymond, Iker Badiola, Serge Evrard, Bruno Villoutreix, Simon Pernot, Géraldine Siegfried, and Abdel Majid Khatib. 2025. *Repression of Apelin Furin Cleavage Sites Provides Antimetastatic Strategy in Colorectal Cancer*. Vol. 17. Springer US.
- Helker, Christian S. M., Jean Eberlein, Kerstin Wilhelm, Toshiya Sugino, Julian Malchow, Annika Schuermann, Stefan Baumeister, Hyouk Bum Kwon, Hans Martin Maischein, Michael Potente, Wiebke Herzog, and Didier Y. R. Stainier. 2020. "Apelin Signaling Drives Vascular Endothelial Cells towards a Pro-Angiogenic State." *ELife* 9:1–44. doi: 10.7554/ELIFE.55589.
- Kleinz, Matthias J., and Anthony P. Davenport. 2005. "Emerging Roles of Apelin in Biology and Medicine." *Pharmacology and Therapeutics* 107(2):198–211. doi: 10.1016/j.pharmthera.2005.04.001.
- Malchow, Julian, Jean Eberlein, Wei Li, Benjamin M. Hogan, Kazuhide S. Okuda, and Christian S. M. Helker. 2024. "Neural Progenitor-Derived Apelin Controls Tip Cell Behavior and Vascular Patterning." *Science Advances* 10(27). doi: 10.1126/sciadv.adk1174.
- Shin, Kyungsoo, Calem Kenward, and Jan K. Rainey. 2018. "Apelinergic System Structure and Function." *Comprehensive Physiology* 8(1):407–50. doi: 10.1002/cphy.c170028.

We thank again the reviewer's supportive comments to improve our manuscript.

Reviewer #1 (Remarks to the Author):

Herd, L. et al. have addressed the major concerns from the three reviewers. Both reviewers #1 and #3 were concerned with the responses of APLNR-circularly permuted GFP probes (APLNR-cpGFP) to endogenous APLNR ligands. To test whether APLNR-cpGFP can monitor the endogenous ligand gradient in vivo, they examined the effect of depletion of Apelin and Apela on the activity of APLNR-cpGFP. In Apelin/Apela CRISPANTS, the APLNR-cpGFP was decreased (new Figure 5e). In addition, they examined the effect of overexpression of APLNR ligands on APLNR-coGFP (new Figure 5g and h). These data indicate that APLNR-cpGFP fluorescence clearly reflects the responses of the probes to endogenous APLNR ligands. Moreover, to address the concern from reviewer #2 about the expression difference of the probes presumably under cell circumstances, they developed new ratiometric biosensors. These new data fortify the usefulness of APLNR conformation biosensors that they developed to monitor APLNR activity in vivo. Given the new results, the authors successfully addressed most of the concerns from the reviewers. For the future study, the authors are encouraged to knock-in the probe in the endogenous loci of zebrafish APLNR to avoid the effect of overexpression of probes using GAL4/UAS system.

The following points should be carefully described.
1. The authors still need to clarify the APLNR-cpGFP biosensor quantification in vivo described in the methods (from line 599). The normalization of fluorescence intensity using DA/PCV is unclear. It is also unclear how the authors additionally normalized the intensity of ISV to compare that of the control and with that of the CRISPANTS (Figure 5).

We thank the Reviewer for these important and legitimate points. Based on previous findings that Apelin signalling is required for the formation of intersegmental vessels (ISVs) but not the dorsal aorta (DA) and posterior cardinal vein (PCV)^{1,2}, we hypothesized that Apelin signalling would be more active in the ISVs than in other vascular beds. To test this hypothesis, we normalized the fluorescence intensity measured by the APLNR-cpGFP biosensor in the ISVs to the fluorescence intensity in the DA/PCV in Figure 5a,c. This approach allowed us to highlight the relative increase in Apelin signalling activity in the ISVs compared to the DA/PCV.

We have clarified this rationale and methodology in the revised manuscript to ensure that the normalization procedure is clearly understood.

We thank the Reviewer for pointing this out. For the CRISPANT experiments, we quantified the APLNR(K235)-cpGFP biosensor fluorescence intensities in the ISVs of *apln* and *apela* CRISPANT embryos, as well as their control siblings. To enable a direct comparison between experimental groups, we normalized the cpGFP fluorescence intensities of each ISV to the mean cpGFP fluorescence intensity of control siblings. This allowed us to assess changes in Apelin signalling activity relative to the baseline control levels.

2. There is no explanation of arrowheads of Figure 3 in the legends.

We thank the Reviewer for this comment. We have now added the description of the arrowheads in the Figure legend.

Reviewer #2 (Remarks to the Author):

The authors have addressed all my comments and I happy to advise publication of the manuscript.

We thank the Reviewer for the positive feedback.

Reviewer #3 (Remarks to the Author):

We thank the Reviewer for the positive feedback.